# Machine learning meets complex networks via coalescent embedding in the hyperbolic space

Alessandro Muscoloni [1], Josephine Maria Thomas[1], Sara Ciucci[1,2], Ginestra Bianconi[3] & Carlo Vittorio Cannistraci [1,4]

Physicists recently observed that realistic complex networks emerge as discrete samples from a continuous hyperbolic geometry enclosed in a circle: the radius represents the node centrality and the angular displacement between two nodes resembles their topological proximity. The hyperbolic circle aims to become a universal space of representation and analysis of many real networks. Yet, inferring the angular coordinates to map a real network back to its latent geometry remains a challenging inverse problem. Here, we show that intelligent machines for unsupervised recognition and visualization of similarities in big data can also infer the network angular coordinates of the hyperbolic model according to a geometrical organization that we term "angular coalescence." Based on this phenomenon, we propose a class of algorithms that offers fast and accurate "coalescent embedding" in the hyperbolic circle even for large networks. This computational solution to an inverse problem in physics of complex systems favors the application of network latent geometry techniques in disciplines dealing with big network data analysis including biology, medicine, and social science.

[1] Biomedical Cybernetics Group, Biotechnology Center (BIOTEC), Center for Molecular and Cellular Bioengineering (CMCB), Center for Systems Biology Dresden (CSBD), Department of Physics, Technische Universität Dresden, Tatzberg 47/49, 01307 Dresden, Germany. [2] Lipotype GmbH, Tatzberg 47, 01307 Dresden, Germany. [3] School of Mathematical Sciences, Queen Mary University of London, London, E1 4NS, UK. [4] Brain Bio-Inspired Computing (BBC) Lab, IRCCS Centro Neurolesi "Bonino Pulejo", Messina, 98124, Italy. Alessandro Muscoloni and Josephine Maria Thomas contributed equally to this work. Correspondence and requests for materials should be addressed to C.V.C. (email: kalokagathos.agon@gmail.com)

Significant progress has been achieved in the last 20 years in unveiling the universal properties of complex networks. Nevertheless, the characterization of the large variety of real network structures, which are originating from the "Big Data explosion," remains an important challenge of network science. Network geometry aims at making a paradigmatic shift in our understanding of complex network structures by revealing their hidden metric[1–10]. This field has a large number of applications ranging from brain networks[7] to routing packets in the Internet[11, 12]. In this context, there is increasing evidence that the hidden metric of many complex networks is hyperbolic[13]. Examples of recent research topics are the development of tools to generate hyperbolic networks[14, 15], the measurement of the hyperbolicity of complex networks[16, 17], the analysis of its impact on traffic congestion[18, 19] and on link prediction[20], the characterization of network properties in terms of the parameters of hyperbolic network models[21], and the study of time-varying control systems with hyperbolic network structure[22]. However, the science that studies and designs algorithms to reveal and to test the latent geometry[23] of real complex networks, is in its dawning.

The popularity-similarity-optimization (PSO) model suggests that real networks have a congruous geometrical representation in a hyperbolic space, where each network node is mapped according to the angular and the radial coordinates of a polar system[1]. On one hand, node similarities are related with the angular distances in the hyperbolic space: the higher the similarity between two nodes, the closer their angular coordinates. On the other hand, the node degree is related with the intrinsic popularity of the node: the higher the node degree, the higher its popularity in the network and the lower its radial coordinate in the hyperbolic space. Recently, further variants of the PSO model have been proposed in order to produce hyperbolic synthetic networks with soft communities[24] or with a desired community structure[25].

Manifold machine learning for unsupervised nonlinear dimensionality reduction is an important subclass of topological machine learning algorithms. They learn nonlinear similarities/ proximities (that can be also interpreted as dissimilarities/distances) between points (samples) distributed over a hidden manifold in a multidimensional feature space, in order to preserve, embed (map) and visualize them in a two-dimensional reduced space[26]. They are inspired by a three-step procedure. First, they approximate the shape of the hidden manifold reconstructing a nearest-neighborhood graph between the points in the high-dimensional space. Second, they use the reconstructed network to estimate pairwise topological similarities (or distances) between the points that lie on the manifold, and store these nonlinear estimations in a kernel (or distance matrix). In a third and last step, they apply a matrix decomposition to the kernel to perform dimensionality reduction, usually in a space of two dimensions. If the network is already given in the form of an unweighted adjacency matrix, the same algorithm works neglecting the first step and thus, in practice, performs a network embedding that preserves the node similarities. These methods are already used in network biology for instance to predict node similarities in protein interaction networks[4, 8], therefore it was likely for us to envisage their usage for network embedding in the hyperbolic space.

Here we show that, adopting topological-based machine learning for nonlinear dimension reduction, the node angular coordinates of the hyperbolic model can be directly approximated in the two- or three-dimensional embedding space according to a persistent node aggregation pattern, which we term "angular coalescence." Based on this phenomenon, we propose a class of algorithms that offers fast (time complexity approximately $O(N^2)$, with $N$ indicating the network node size) and accurate "coalescent

embedding" in the two- or three-dimensional hyperbolic space even for large unweighted and weighted networks. This discovery paves the way for the application of network latent geometry

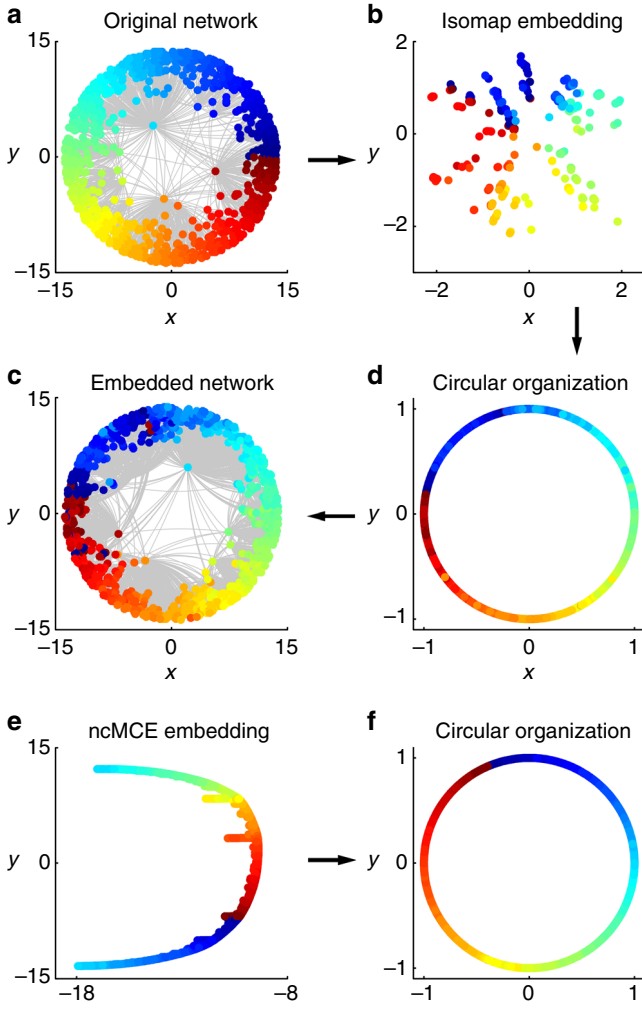

**Fig. 1** Coalescent embedding. **a** We show the original synthetic network generated by the PSO model in the hyperbolic space. **b** The Isomap algorithm (ISO), which is the progenitor of manifold techniques, starting from the unweighted adjacency matrix offers an embedding of the network nodes that is organized according to a circular pattern that follows the angular coordinates of the original PSO model. We made different trials using other synthetic networks, and this circular pattern is mainly preserved if the kernel is centered or if the kernel is not centered and the first dimension is neglected (see "Methods" section for details). This makes sense because the operation of kernel centering puts the origin of the reduced space at the center of the points in a multidimensional space and thus at the center of the manifold. Since the node points lie on the hyperbolic disk, the embedding places the origin approximatively at the center of the disk. **d** The nodes are projected over a circumference and adjusted equidistantly according to the step 3.2 of the algorithm described in "Methods" section. **c** The radial coordinates are given according to Eq. (4). **e** A different pattern is obtained for an algorithm named ncMCE. The circular pattern is linearized and the nodes are ordered along the second dimension of embedding according to their similarities (here the kernel is noncentered and the first dimension of embedding should be neglected, see "Methods" section). **f** If we accommodate the node points on the circumference following the same ordering as the second dimension of embedding, we can again recover an unbroken circular pattern that resembles the angular coordinates of the original PSO model

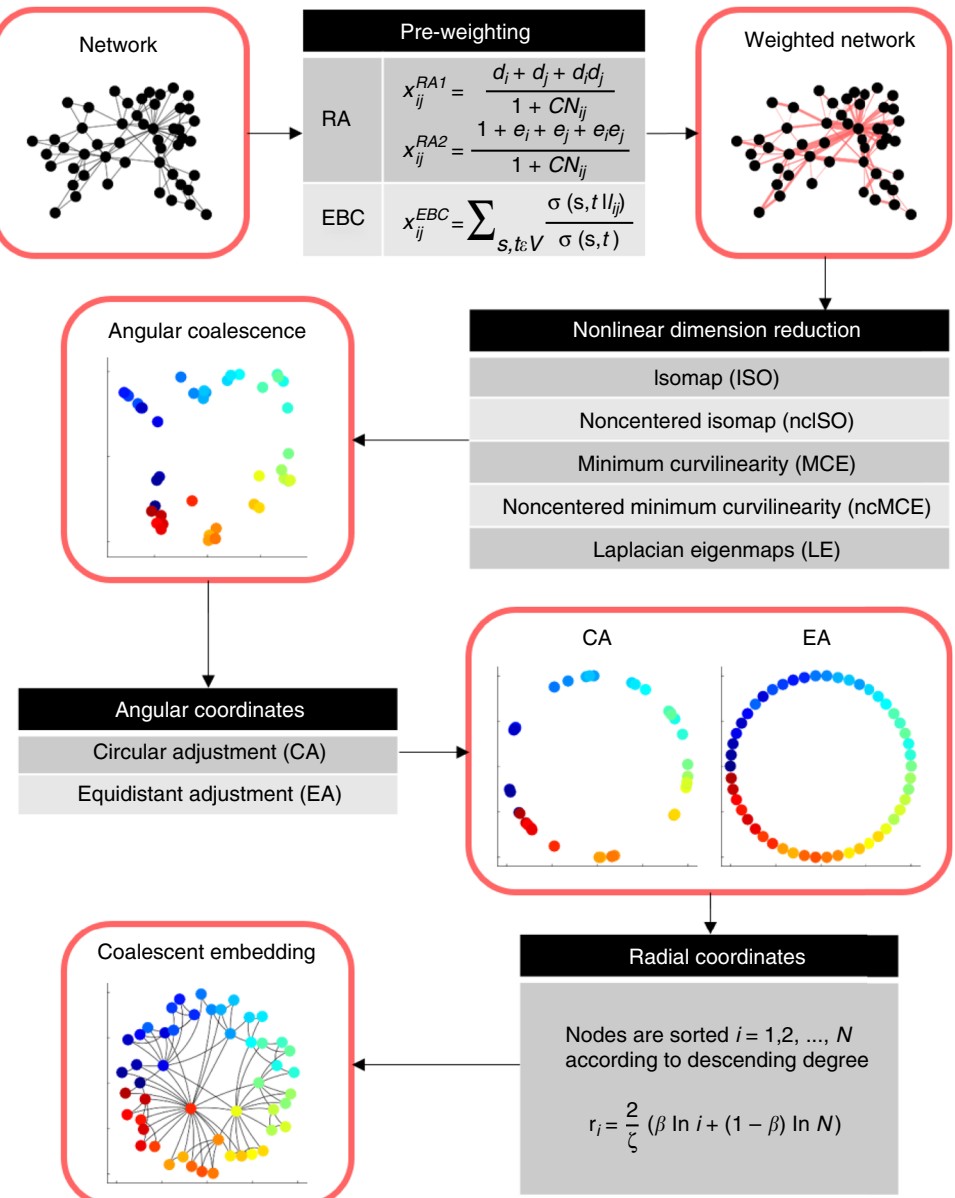

**Fig. 2** Flow chart of the coalescent embedding algorithm. The algorithmic steps (grayscale squares) and the intermediate input/output (rounded red squares) of the coalescent embedding algorithm are illustrated. Each algorithmic step reports all the possible variants. The example network has been generated by the PSO model with parameters $N = 50$, $m = 2$, $T = 0.1$, $\gamma = 2.5$. We applied the $RA_1$ pre-weighting rule and the ISO dimension reduction technique. The colors of the embedded nodes are assigned according to their angular coordinates in the original PSO network. Description of the variables in the mathematical formulas: $x_{ij}$ value of $(i, j)$ link in adjacency matrix $x$; $d_i$ degree of node $i$; $e_i$ external degree of node $i$ (links neither to $CN_{ij}$ nor to $j$); $CN_{ij}$ common neighbors of nodes $i$ and $j$; $V$ set of nodes; $s,t$ any combination of network nodes in $V$; $\sigma(s, t)$ number of shortest paths $(s,t)$; $\sigma(s, t|l_{ij})$ number of shortest paths $(s, t)$ through link $l_{ij}$; $N$ number of nodes; $\zeta = \sqrt{-K}$, we set $\zeta = 1$; $K$ curvature of the hyperbolic space; $\beta = \frac{1}{\gamma-1}$ popularity fading parameter; $\gamma$ exponent of power-law degree distribution. Details on each step are provided in the respective "Methods" sections

techniques in many disciplines dealing with big network data analysis including biology, medicine, and social science.

## Results

**Coalescent embedding**. In this study, we selected a representative group of nonlinear topological-based unsupervised dimensionality reduction approaches among the ones with the highest performance. Three manifold-based: Isomap (ISO)[27], noncentered Isomap (ncISO)[8], and Laplacian eigenmaps (LE)[28]. Two minimum-curvilinearity-based: minimum curvilinear embedding (MCE)[8, 26] and noncentered minimum curvilinear embedding

(ncMCE)[8]. An important note for practical applications is that these approaches are unsupervised (node or edge labels are not required) and parameter-free (external tuning setting of algorithms' parameters is not required).

In Fig. 1b–d, we show the embedding provided by the Isomap algorithm (ISO), the progenitor of the manifold dimension reduction techniques, starting from the unweighted adjacency matrix of a PSO network. The nodes are organized according to a circular pattern (Fig. 1b), which follows the angular coordinates of the original PSO model. For an algorithm named noncentered minimum curvilinear embedding (ncMCE)[8] (Fig. 1e, f), the circular pattern is linearized (Fig. 1e) and the nodes are ordered

along the second dimension of embedding according to their similarities. If we accommodate the node points on the circumference following the same ordering as the second dimension of embedding (Fig. 1e), we can again recover an unbroken circular pattern (Fig. 1f) that resembles the angular coordinates of the original PSO model. The ability of ncMCE and minimum-curvilinearity-based algorithms to learn, unfold, and linearize along just one dimension an intrinsic nonlinear (circular) pattern is discussed in details in the "Methods" section. However, here we clarify that minimum-curvilinearity-based algorithms compress the information in one unique dimension because they learn nonlinear similarities by means of the minimum spanning tree (MST), providing a hierarchical-based mapping that is fundamentally different from the manifold-based of ISO.

The rationale of our approach is all contained in these simple insights. We embedded hyperbolic networks adopting different combinations of network similarities and matrix decompositions and we reached the same consistent finding: the arising in the two-dimensional embedding space of a common node aggregation pattern, which we named "angular coalescence," and that was circularly or linearly ordered according to the angular coordinates of the hyperbolic model. This represents the first important result of our study. The term "angular coalescence" is proposed to indicate that the individual nodes aggregate together (from the Latin verb coalēscō: to join, merge, amalgamate single elements into a single mass or pattern) forming a pattern that is progressively ordered along the angular dimension. Consequently, we decided to coin the expression "coalescent embedding" to indicate the class of algorithms that exhibit angular coalescence in the two-dimensional network embedding. In our case, we detected the angular coalescence phenomenon as embedding result of topological-based machine learning for nonlinear unsupervised dimension reduction. Indeed, the evidence that even MCE and ncMCE, which are not manifold-based but hierarchical-based, are able to exhibit coalescent embedding may theoretically suggest that this is an "epiphenomenon" that in general characterizes topological-based machine learning for nonlinear dimension reduction when applied to this task.

Given the first results, we propose to adopt these machine learning techniques to perform two-dimensional "structural network imaging," which could be figuratively envisaged as a sort of in silico imaging technique (such as X-ray or MRI is for condensed matter) for 2D reconstruction and visualization of the hidden manifold shape from which the structural organization of a complex network emerges.

In the "Methods" section, we propose a general algorithm—based on the angular coalescence principle—for network embedding in the hyperbolic space. In Fig. 2, a flow chart is reported, where both the algorithmic steps and the intermediate input/output are highlighted. In order to build a general algorithm, we started by noticing that the problem to compute the embedding on an unweighted adjacency matrix would be simplified by having a "good guess" of the edge weights that suggest the connectivity geometry. Thus, there was a clear margin to improve the coalescent embedding performance by pre-weighting the network links using a convenient strategy to approximate distances between the connected nodes. We devised two different pre-weighting strategies. The first approach—which we called the repulsion–attraction rule (RA)—assigns an edge weight adopting only the local information related to its adjacent nodes (neighborhood topological information). The idea is that adjacent nodes with a high external degree (where the external degree is computed considering the number of neighbors not in common) should be geometrically far because they represent hubs without neighbors in common, which—according to the theory of navigability of complex networks presented by Boguñá et al.[2]—

tend to dominate geometrically distant regions: this is the repulsive part of the rule. On the contrary, adjacent nodes that share a high number of common neighbors should be geometrically close because most likely they share many similarities: this is the attractive part of the rule. Thus, the RA (see Eqs. (1) and (2) for two alternative mathematical formulations) is a simple and efficient approach that quantifies the trade-off between hub repulsion and common neighbors-based attraction. Supplementary Fig. 1 gives a visual example about how the RA pre-weighting rule is improving the angular coalescence effect with respect to Fig. 1, where the same methods are adopted without pre-weighting. Since it might be argued that the repulsion between high external degree nodes implied by the RA rule is in contrast with the existence of rich clubs, in Supplementary Discussion we comment the rich clubness of the PSO networks (Supplementary Fig. 25), and why this does not affect the RA pre-weighting efficiency. Although inspired by the same rationale, the second strategy makes, instead, a global-information-based pre-weighting of the links, using the edge betweenness centrality (EBC) to approximate distances between nodes and regions of the network. EBC is indeed a global topological network measure, which expresses for each edge of the network a level of centrality, and the assumption is that central edges are bridges that tend to connect geometrically distant regions of the network, while peripheral edges tend to connect nodes in the same neighborhood. We let notice that if a weighted network is given, where the weights suggest distances between connected nodes, these can be directly adopted rather than approximated by the pre-weighting techniques.

Furthermore, we were not convinced that preserving the angular distances between nodes adjacent in the angular coordinates was the best strategy. Most likely their reciprocal angular distances were affected by short-range angular noise. Thus, we devised a strategy to reorganize the nodes on the circumference that we called equidistant adjustment (EA): the nodes are equidistantly reorganized along the angular coordinates of the circumference according to their original order learned by the coalescent embedding. Figure 2 displays a didactic example of the difference between the circular and equidistant adjustment.

The several variants of coalescent embedding algorithms, characterized by the different pre-weightings and angular adjustments, have been tested in various evaluation frameworks using both synthetic and real networks, and their performance has been compared to state-of-the-art methods for hyperbolic embedding. The next sections will report the results obtained together with a discussion of the main achievements.

**Evaluations of mapping accuracy in popularity-similarity-optimization synthetic networks**. In order to test the performance of the hyperbolic embedding methods, synthetic networks have been generated with the PSO model, ranging over several combinations of the parameters. Figure 3 reports the results of the best dimension reduction methods for the first evaluation framework. Here the performance was evaluated as Pearson correlation between all the pairwise hyperbolic distances of the network nodes (we called such correlation: HD-correlation) in the original PSO model and in the reconstructed hyperbolic space. The plots report the average correlation over the 100 synthetic networks that have been generated for each different PSO model parameter combination. It is evident that the coalescent embedding techniques pre-weighted with RA and adjusted according to EA are outperforming HyperMap[29], HyperMap-CN[30], and LPCS[31] that are the state-of-the-art, and this is the second key discovery of our study. RA performed similarly to EBC, and in general both the pre-weighting strategies are effective

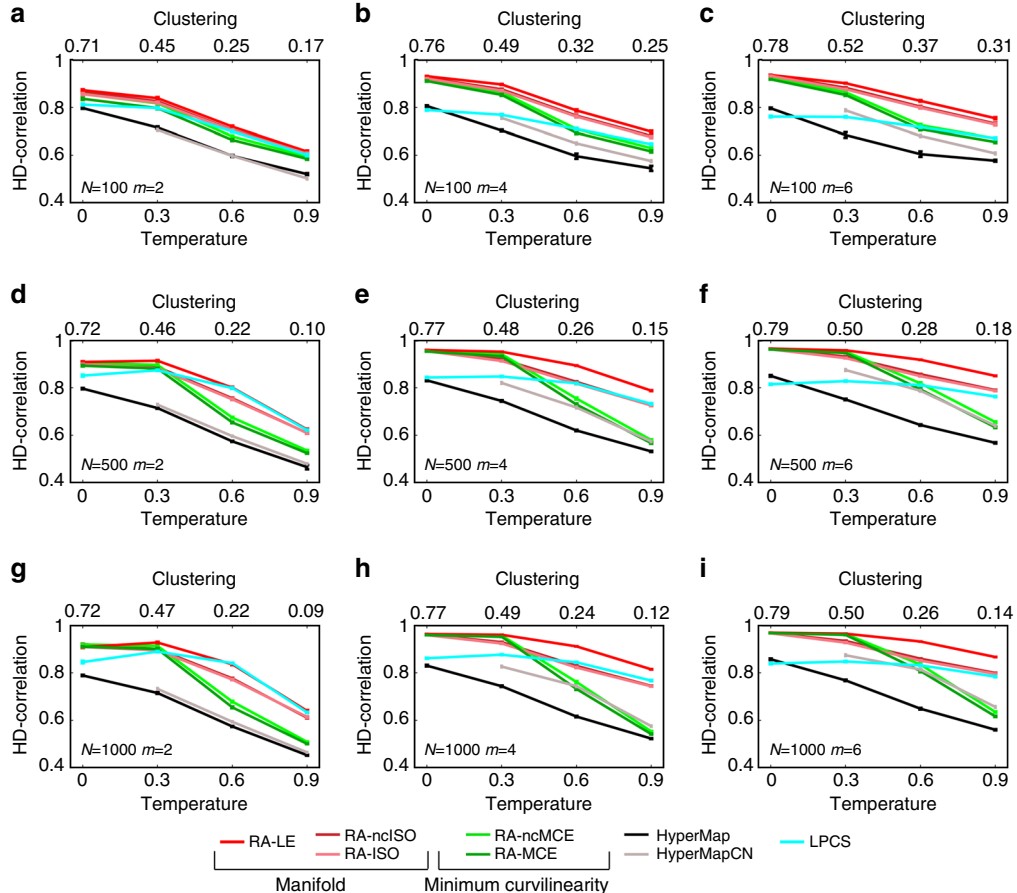

**Fig. 3** HD-correlation on popularity-similarity-optimization synthetic networks. **a–i** To validate the above-mentioned techniques, we generated 100 different synthetic networks for each combination of tuneable parameters of the PSO model (temperature $T$, size $N$, half of average degree $m$, power-law degree distribution exponent $\gamma$). Supplementary Fig. 24 offers an idea of the topological diversity of the synthetic networks generated fixing $\gamma = 2.5$ and tuning the other parameters, Supplementary Fig. 25 reports an analysis of the rich clubness of the networks, commented in Supplementary Discussion. In the results presented in the figures of this article, we used $\gamma = 2.5$, but we also ran the simulations for $\gamma = 2.25$ and 2.75, and the differences were negligible (results not shown). Here, the performance was evaluated as Pearson correlation between all the pairwise hyperbolic distances of the network nodes in the original PSO model and in the reconstructed hyperbolic space (HD-correlation). The plots report the average correlation and the standard error over the 100 synthetic networks that have been generated for each different parameter combination. The value one indicates a perfect correlation between the nodes' hyperbolic distances in the original and reconstructed hyperbolic space. The plots show the results of different methods when both RA and EA are applied. The methods without EA are plotted in Supplementary Fig. 7. For each subplot, the value of HyperMap-CN for $T = 0$ is missing because the original code assumes $T > 0$

(Supplementary Figs. 2–6). However, RA is computationally more efficient because it is a local approach (see "Methods" section for details about the complexity). Obviously, all the methods reduce their performance for increasing temperature (reduced clustering), because the networks assume a more "random" structure.

Another alluring result, pointing out a very subtle problem, is that without EA all the techniques significantly reduce the performance, as it is shown in Supplementary Fig. 7. Looking at Fig. 3 and the Supplementary Figs. 2–6, EA makes a difference especially for low temperatures (high clustering), while for high temperatures its improvement is vanishing. This is particularly evident for LE that in Supplementary Fig. 7 (where EA is not applied) at low temperatures has a significantly worse performance compared to Fig. 3, where EA is applied. Imposing an equidistant adjustment might be counterintuitive, but our simulations suggest that this sub-optimal strategy is better than passively undergo the short-range angular embedding uncertainty. On the other hand, once the temperature is increased, the overall angular embedding uncertainty also increases and the techniques are less efficient to recover the node order. In practice,

for high temperatures, the overall noise overcomes the short-range noise and the EA reduces its effectiveness.

In Fig. 4j-r, we repeated the same evaluation of Fig. 3, but we adopted a different measure called Concordance score (C-score). The C-score can be interpreted as the proportion of node pairs for which the angular relationship in the inferred network corresponds to the angular relationship in the original network (see "Methods" section). Basically, this score provides a quantitative evaluation of the scatter plots in Fig. 4a-i, where the alignment between inferred and original angular coordinates is visually compared. A C-score of 0 indicates total misalignment, while 1 indicates perfect alignment. The results in Fig. 4 confirm that our methods outperform, especially for low temperatures, the state-of-the-art techniques also in recovering a good angular alignment of the nodes. Supplementary Figs. 8–17 show the scatter plots for all the other methods and temperatures. The scatter plots visually highlight that the correlation between real and inferred angular coordinates decreases for increasing temperatures. However, it is evident that the proposed techniques are able to provide quite accurate alignments even for middle

temperatures. Noticeably, the coalescent embedding-based algorithms combine important performance improvement with a spectacular speed up in respect to HyperMap (Fig. 5 and

Supplementary Fig. 18). They can even embed large networks of 10,000 nodes in less than 1 min and 30,000 nodes in few minutes (Fig. 5), whereas HyperMap requires more than 3 h for small

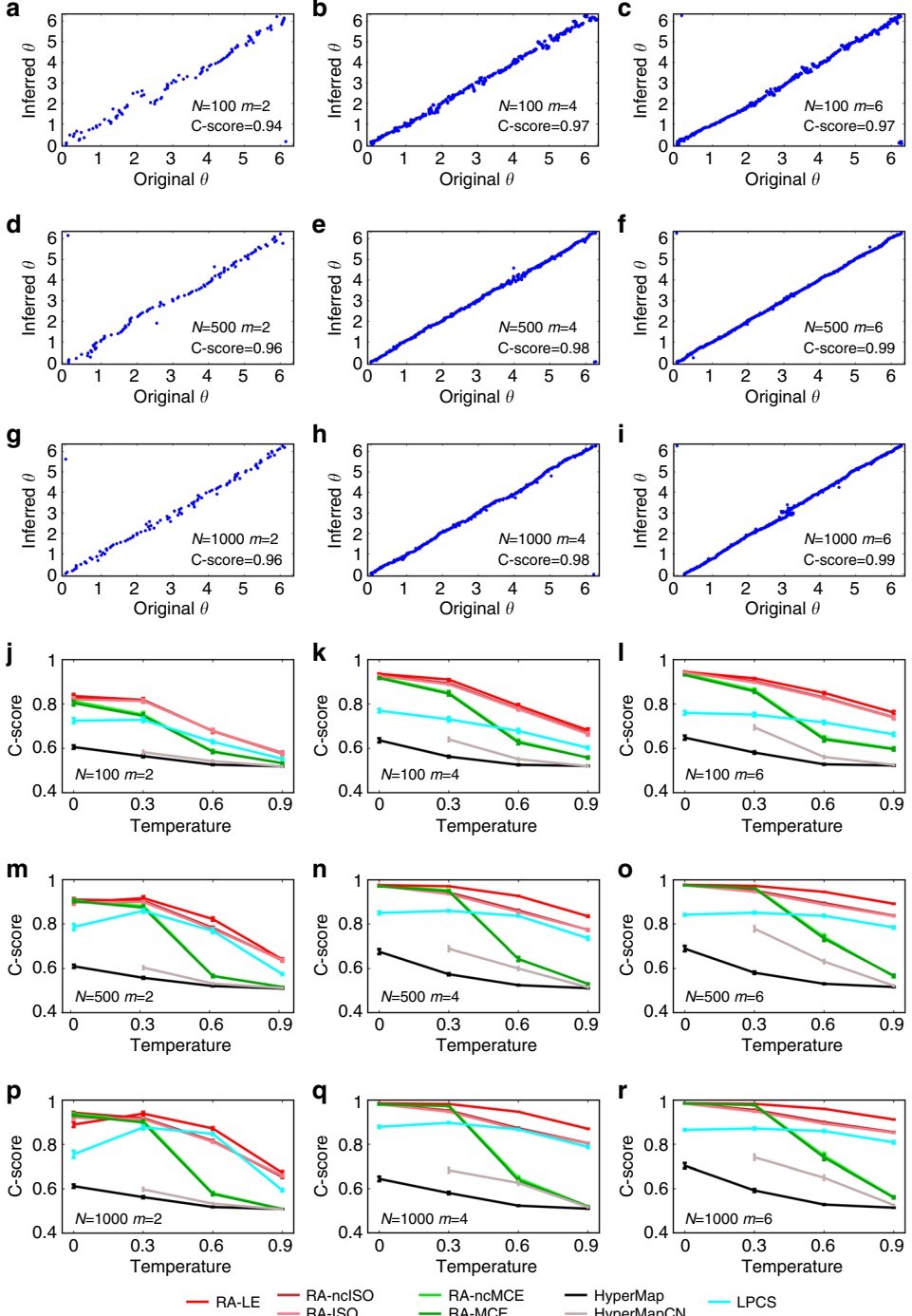

**Fig. 4** Angular coordinates comparison and C-score on popularity-similarity-optimization synthetic networks. **a-i** For all the combinations of the PSO parameters $N$ (size) and $m$ (half of average degree), we chose among the synthetic networks embedded with RA-MCE-EA the ones with the best C-score, which had always temperature $T = 0$. For these networks, we plotted the aligned inferred angular coordinates against the original angular coordinates ($\theta$). The alignment was done in the following manner: we applied 360 rotations of one degree both to the inferred coordinates as they are and to the inferred coordinates obtained arranging the nodes in the opposite clock direction. Then from these resulting 720 alternatives of the inferred angular coordinates, we chose the one that maximizes the correlation with the original angular coordinates, in order to guarantee the best alignment. The alignment does not change the C-score, which represents the percentage of node pairs in the same circular order in the original and inferred networks (see "Methods" section for details). Similar plots for the other coalescent embedding methods and temperature values can be found in Supplementary Figs. 8-17. **j-r** The plots report the average C-score and the standard error over the 100 synthetic networks that have been generated for each different parameter combination. There are no separate plots for the methods with and without EA since this adjustment affects the distances but not the circular ordering, therefore it does not change the C-score. For each subplot, the value of HyperMap-CN for $T = 0$ is missing because the original code assumes $T > 0$

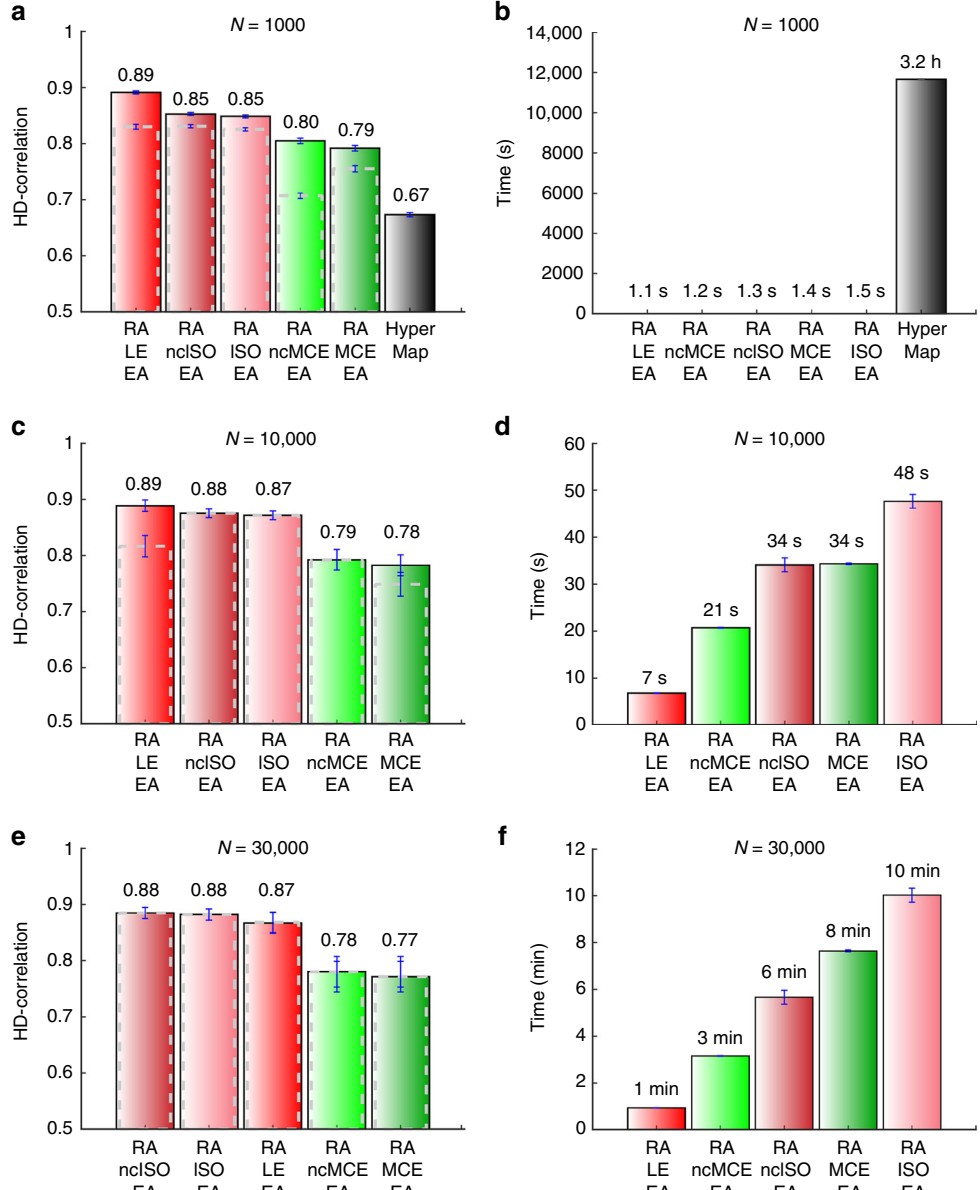

**Fig. 5** Comparison of HD-correlation and time on small and large-size popularity-similarity-optimization synthetic networks. **a**, **c**, **e** Average performance and standard error, measured as HD-correlation, for all PSO networks of sizes $N = 1000$, $N = 10,000$, and $N = 30,000$, respectively. Averages are taken over the parameters $m$ (half of the mean node degree) and temperature $T$. **b**, **d**, **f** Average computation times for the PSO networks of sizes $N = 1000$, $N = 10,000$, and $N = 30,000$, respectively. Again, averages are taken over the parameters $m$ (half of the mean node degree) and temperature $T$. Considering the average performance in all the simulations on 1000 nodes networks **a**, coalescent embedding approaches achieved a performance improvement of more than 30% in comparison to HyperMap, requiring only around one second versus more than three hours of computation time. Similar performance results are confirmed for the networks of sizes $N = 10,000$ and $N = 30,000$ with an execution time still in the order of minutes for the biggest networks. The comparison to HyperMap was not possible due to its long running time. The dashed gray bins represent the HD-correlation of the respective non-EA variants, suggesting that their performance tends to the EA variants for larger PSO networks

networks of just 1000 nodes. It is important to underline that, in addition to the remarkable scaling of the computational time (see "Methods" section for details about the complexity), the high-correlation values are also preserved for larger networks.

**Greedy routing performance in synthetic and real networks.** Another important characteristic that can be studied in a network embedded in a geometrical space is its navigability. The network is considered navigable if the greedy routing (GR) performed using the node coordinates in the geometrical space is efficient[2].

In the GR, for each pair of nodes, a packet is sent from the source to the destination and each node knows only the address (coordinates) of its neighbors and the address of the destination, which is written in the packet. In the GR procedure adopted[29], at each hop the packet is forwarded from the current node to its neighbor at the lowest hyperbolic distance from the destination and it is dropped when a loop is detected. The efficiency is evaluated according to the GR-score (see "Methods" section for details), which assumes values between 0, when all the routings are unsuccessful, and 1, when all the packets reach the destination through the shortest path. Supplementary Fig. 19 compares the

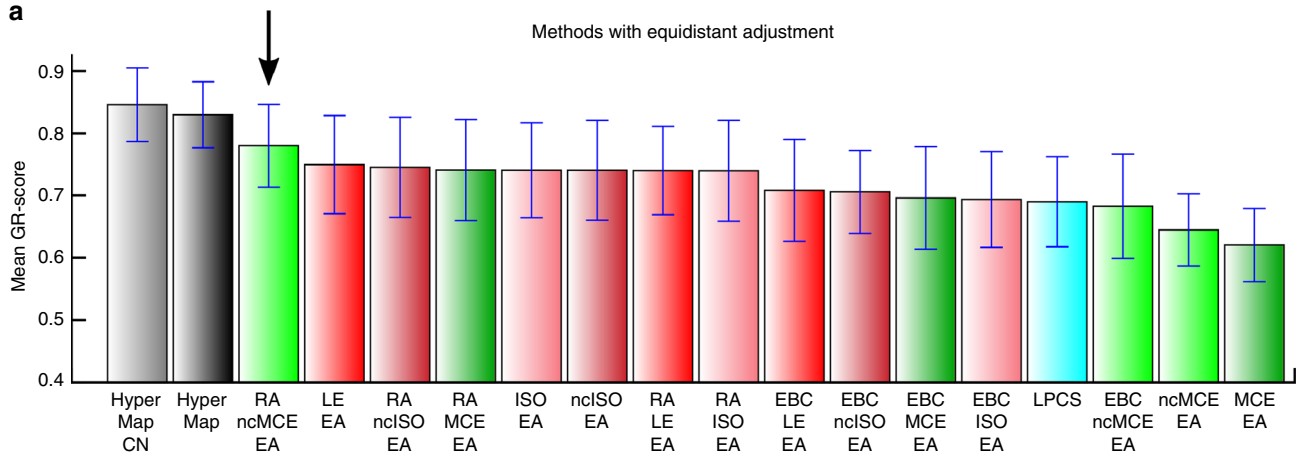

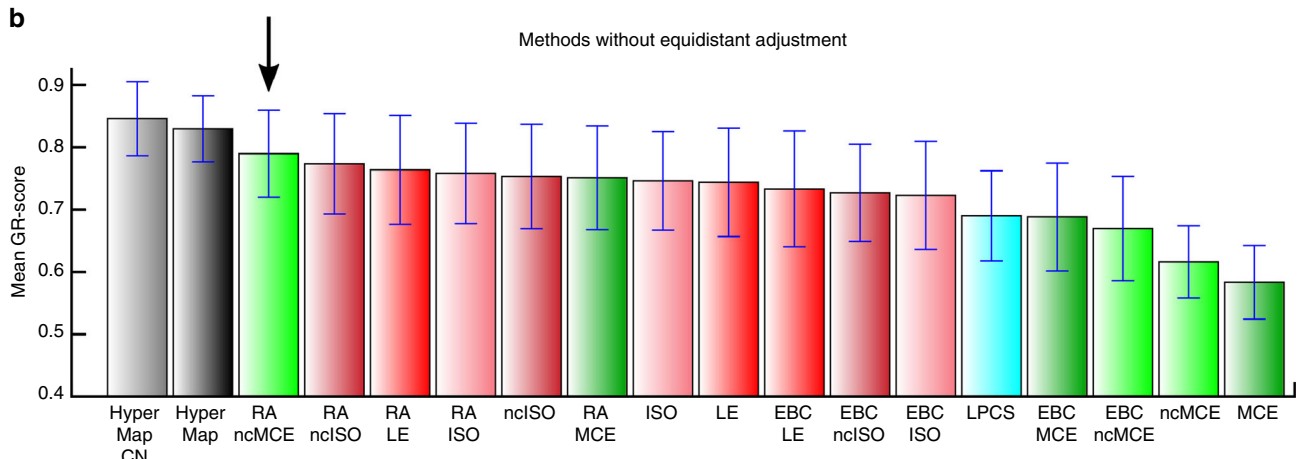

**Fig. 6** Greedy routing (GR) on real networks. The eight real networks whose statistics are reported in Table 1 have been mapped using the hyperbolic embedding techniques and the greedy routing in the geometrical space has been evaluated. The barplot report for each method the mean GR-score and standard error over the networks. The GR-score is a metric to evaluate the efficiency of the greedy routing, which assumes values between 0, when all the routings are unsuccessful, and 1, when all the packets reach the destination through the shortest path (see "Methods" section for details). Both the EA (**a**) and non-EA (**b**) variants are reported, in order to check whether the equidistant adjustment might affect the navigability. A black arrow points the coalescent embedding algorithm RA-ncMCE that offers the best performance regardless the use of node angular adjustment. The mean GR-score of RA-ncMCE is not statistically different from the one of the HyperMap-based algorithms (permutation test *p* value >0.2 in all the pairwise comparisons)

performance of the hyperbolic embedding methods as mean GR-score over all the PSO networks of Fig. 3. While mean GR-score on eight real networks (whose statistics are reported in Table 1) are shown in Fig. 6. The first fact to notice is that the PSO network as synthesized with its original coordinates is the most navigable network. Second, HyperMap-based algorithms obtained the highest GR-score among the hyperbolic embedding algorithms, followed by the coalescent embedding technique RA-ncMCE, which turns out to be the best both considering EA and non-EA versions. However, the mean GR-score of RA-ncMCE in Fig. 6 is not statistically different from the one of the HyperMap-based algorithms (permutation test *p* value >0.2 in all the pairwise comparisons), therefore their performance is comparable on real networks of Fig. 6. This is an impressive result and we will now explain the reason. The success of the GR is very sensitive to the fact that connected nodes are mapped close in the geometrical space and disconnected nodes far apart. In fact, mapping disconnected nodes close in the geometrical space is likely to cause the routing of packets into wrong paths. In the original PSO network, nodes are connected with probability inversely proportional to their hyperbolic distance[1], therefore connected nodes tend to be close and disconnected nodes faraway by construction,

which explains the high navigability of the networks generated with the PSO model. The reason why HyperMap methods offer the best GR performance is that—during maximum likelihood estimation procedure—they iteratively adjust both the angular and radial coordinates of the nodes using an objective function that is maximized if connected nodes are at low hyperbolic distance and disconnected nodes are at high hyperbolic distance[29]. The reason why coalescent embedding techniques offer a GR performance that is inferior to HyperMap methods is that they put connected nodes close and disconnected far only in the angular coordinates and not directly in the hyperbolic space, where instead the GR navigation occurs. In brief, coalescent embedding optimizes angular distances in order to put connected points close and disconnected far, while HyperMap optimizes the hyperbolic distances. Therefore, the results obtained by RA-ncMCE in GR are impressive considering that for this method only angular coordinates contribute to the organization of the points in the hyperbolic space, and that despite this significant limitation RA-ncMCE performances on real networks are comparable to the ones of HyperMap methods. This finding is promising since further algorithms might also be designed to embed directly in the hyperbolic space instead of inferring exclusively

**Table 1 Community detection on real networks with Louvain algorithm**

| Method | Karate $N = 34$ $E = 78$ $m = 2.29$ $T = 0.43$ $\gamma = 2.12$ $N_c = 2$ | Opsahl 8 $N = 43$ $E = 193$ $m = 4.49$ $T = 0.43$ $\gamma = 8.20$ $N_c = 7$ | Opsahl 9 $N = 44$ $E = 348$ $m = 7.91$ $T = 0.32$ $\gamma = 5.92$ $N_c = 7$ | Opsahl 10 $N = 77$ $E = 518$ $m = 6.73$ $T = 0.35$ $\gamma = 5.06$ $N_c = 4$ | Opsahl 11 $N = 77$ $E = 1088$ $m = 14.13$ $T = 0.28$ $\gamma = 4.87$ $N_c = 4$ | Polbooks $N = 105$ $E = 441$ $m = 4.20$ $T = 0.51$ $\gamma = 2.62$ $N_c = 3$ | Football $N = 115$ $E = 613$ $m = 5.33$ $T = 0.60$ $\gamma = 9.09$ $N_c = 12$ | Polblogs $N = 1222$ $E = 16,714$ $m = 13.68$ $T = 0.68$ $\gamma = 2.38$ $N_c = 2$ | Mean | % Impr. |
|---|---|---|---|---|---|---|---|---|---|---|
| EBC-ncISO-EA | **1.00** | 0.57 | 0.47 | 1.00 | 0.93 | 0.59 | 0.90 | 0.68 | 0.77 | **+13.2** |
| RA-MCE-EA | **0.83** | 0.51 | 0.47 | 1.00 | **0.96** | 0.57 | 0.82 | 0.67 | 0.73 | +7.4 |
| RA-ncMCE-EA | 0.73 | 0.55 | 0.47 | 1.00 | **1.00** | 0.57 | 0.83 | 0.67 | 0.73 | +7.4 |
| EBC-MCE-EA | 0.83 | 0.47 | 0.41 | 1.00 | 0.96 | 0.57 | 0.90 | 0.62 | 0.72 | +5.9 |
| EBC-ncMCE-EA | 0.88 | 0.46 | 0.41 | 1.00 | 0.96 | 0.57 | 0.85 | 0.62 | 0.72 | +5.9 |
| EBC-ISO-EA | 0.83 | 0.42 | 0.47 | 1.00 | 0.89 | 0.59 | 0.88 | 0.66 | 0.72 | +5.9 |
| LPCS | 0.83 | 0.49 | 0.41 | 1.00 | 0.96 | 0.55 | 0.87 | 0.67 | 0.72 | +5.9 |
| ncMCE-EA | 0.73 | 0.47 | 0.47 | 1.00 | 0.96 | 0.57 | 0.89 | 0.62 | 0.71 | +4.4 |
| RA-LE-EA | 0.67 | 0.48 | 0.53 | 1.00 | 0.92 | 0.56 | 0.82 | 0.70 | 0.71 | +4.4 |
| RA-ncISO-EA | 0.67 | 0.54 | 0.42 | 1.00 | 0.92 | 0.56 | 0.86 | 0.67 | 0.70 | +2.9 |
| ncISO-EA | 0.73 | 0.50 | 0.41 | 1.00 | 0.88 | 0.54 | 0.87 | 0.66 | 0.70 | +2.9 |
| EBC-LE-EA | 0.85 | 0.42 | 0.41 | 0.96 | 0.92 | 0.56 | 0.85 | 0.62 | 0.70 | +2.9 |
| MCE-EA | 0.64 | 0.47 | 0.47 | 0.96 | 0.92 | 0.55 | 0.86 | 0.62 | 0.69 | +1.5 |
| *unweighted* | *0.46* | *0.55* | *0.41* | *1.00* | *0.96* | *0.50* | *0.93* | *0.64* | *0.68* | *0.0* |
| LE-EA | 0.63 | 0.55 | 0.41 | 1.00 | 0.78 | 0.55 | 0.82 | 0.67 | 0.68 | 0.0 |
| RA-ISO-EA | 0.57 | 0.43 | 0.44 | 1.00 | 0.88 | 0.54 | 0.86 | 0.67 | 0.67 | −1.5 |
| ISO-EA | 0.34 | 0.50 | 0.41 | 0.96 | 0.93 | 0.56 | 0.82 | 0.67 | 0.65 | −4.4 |
| HyperMap | 0.56 | 0.60 | 0.28 | 0.92 | 0.85 | 0.50 | 0.83 | 0.69 | 0.65 | −4.4 |
| HyperMap-CN | 0.55 | 0.47 | 0.41 | 0.93 | 0.79 | 0.54 | 0.79 | 0.70 | 0.65 | −4.4 |

Normalized mutual information (NMI) computed between the ground truth communities and the ones detected by the Louvain algorithm for eight real networks. NMI = 1 indicates a perfect match between the two partitions of the nodes. For each method, the network has been embedded in the hyperbolic space and the embedding coordinates are used to weight the input matrix for the Louvain algorithm: observed links are weighted using the hyperbolic distances between the nodes and non-observed links using the hyperbolic shortest paths (see "Methods" section for details). As a reference, the Louvain algorithm has been run giving in input also the unweighted adjacency matrix, the related row is highlighted in italic. The table contains also some statistics for each network: number of nodes $N$, number of edges $E$, temperature $T$ (inversely related to the clustering coefficient), power-law degree distribution exponent $\gamma$, half of average degree $m$, and number of ground truth communities $N_c$. Due to the higher performance, only the EA methods are here reported, whereas the complete table is shown as Supplementary Table 1. The NMI values highlighted in bold for the Karate and Opsahl_11 networks are the ones whose embedding is shown in Fig. 7. The rightmost column reports the percentage of improvement with respect to the unweighted variant, the best result is highlighted in bold. The results obtained only weighting the observed links are shown in Supplementary Table 5

angular coordinates, as for the moment coalescent embedding is able to do. A digression on the reason why RA-ncMCE is the best performing among the coalescent embedding methods is provided in Supplementary Discussion, together with an analysis on the impact of the equidistant adjustment for GR, reported in Supplementary Fig. 20.

**Community detection on real networks.** Once in possession of fast methods that are able to map complex networks in the hyperbolic space with high precision and to disclose the hidden geometry behind their topologies, several studies can be lead exploiting the geometrical information. The analyses can cover disparate fields like social, communication, transportation, biological, and brain networks. As an example of application, we show how the hyperbolic distances can be used to feed community detection algorithms. Community structure is one of the most relevant features of real networks, and consists in the organization of network nodes into groups within which the connections are dense, but between which connections are sparser. The development of algorithms for detection of such communities is a key topic that has broad applications in network science, for example, in identifying people with similar interests in social networks, functional molecular modules in biological networks, or papers with related topics in citation networks[32]. We modified four approaches for community detection that a recent comparative study[33] has shown to be the best among the state-of-the-art and that accept in input also weighted adjacency matrices: Louvain[34], Infomap[35], Label propagation[36], and Walktrap[37]. We

demonstrate that they can be boosted when applied to the networks weighted according to the hyperbolic distances, which were inferred by some of our coalescent embedding techniques (see "Methods" section for details). In general, our results show that, regardless of the approach used for community detection, ncISO-based and MCE-based coalescent embedding techniques are significantly better than LE-based in this task on real networks (Tables 1 and 2 and Supplementary Tables 1–4).

The improvement obtained for Infomap is moderate but very reliable: indeed, EBC-ncISO-EA allows always (on every network) the improvement or the same performance in respect to standard Infomap (unweighted). The boost obtained for Louvain is remarkable (but less stable), indeed EBC-ncISO-EA, which is also here the best method, offers an overall improvement of +13.2%. In particular, an astonishing performance is obtained for a social network, the Karate Club[38] (Table 1, first column), where the Louvain algorithm based on the EBC-ncISO-EA embedding reaches the perfect community detection—a result that is evident also in the hyperbolic space visualization (Fig. 7a)—whereas the unweighted Louvain, Infomap, Label propagation, and Walktrap algorithms on the same network attain a mediocre performance. The Karate network represents the friendship between the members of a university Karate club in the United States: communities are formed by a split of the club into two parts, each following one trainer. This is a valuable pedagogic result, indeed to the best of our knowledge it is the first time that the communities present in the Karate network are perfectly untangled by means of the Louvain algorithm (which is ineffective without the "geometrical" boost of the coalescent

**Table 2 Community detection on real networks with Infomap algorithm**

| Method | Karate | Opsahl 8 | Opsahl 9 | Opsahl 10 | Opsahl 11 | Polbooks | Football | Polblogs | Mean | % Impr. |
|---|---|---|---|---|---|---|---|---|---|---|
| | $N = 34$ | $N = 43$ | $N = 44$ | $N = 77$ | $N = 77$ | $N = 105$ | $N = 115$ | $N = 1222$ | | |
| | $E = 78$ | $E = 193$ | $E = 348$ | $E = 518$ | $E = 1088$ | $E = 441$ | $E = 613$ | $E = 16{,}714$ | | |
| | $m = 2.29$ | $m = 4.49$ | $m = 7.91$ | $m = 6.73$ | $m = 14.13$ | $m = 4.20$ | $m = 5.33$ | $m = 13.68$ | | |
| | $T = 0.43$ | $T = 0.43$ | $T = 0.32$ | $T = 0.35$ | $T = 0.28$ | $T = 0.51$ | $T = 0.60$ | $T = 0.68$ | | |
| | $\gamma = 2.12$ | $\gamma = 8.20$ | $\gamma = 5.92$ | $\gamma = 5.06$ | $\gamma = 4.87$ | $\gamma = 2.62$ | $\gamma = 9.09$ | $\gamma = 2.38$ | | |
| | $N_c = 2$ | $N_c = 7$ | $N_c = 7$ | $N_c = 4$ | $N_c = 4$ | $N_c = 3$ | $N_c = 12$ | $N_c = 2$ | | |
| EBC-ncISO-EA | 0.68 | 0.75 | 0.47 | 1.00 | 1.00 | 0.54 | 0.92 | 0.53 | 0.74 | +4.2 |
| ncMCE-EA | 0.68 | 0.74 | 0.47 | 1.00 | 0.93 | 0.50 | 0.92 | 0.52 | 0.72 | +1.4 |
| *unweighted* | *0.55* | *0.69* | *0.47* | *1.00* | *1.00* | *0.52* | *0.92* | *0.52* | *0.71* | *0.0* |
| EBC-MCE-EA | 0.68 | 0.55 | 0.53 | 1.00 | 0.96 | 0.52 | 0.93 | 0.52 | 0.71 | 0.0 |
| EBC-ncMCE-EA | 0.58 | 0.55 | 0.53 | 1.00 | 1.00 | 0.52 | 0.93 | 0.52 | 0.70 | −1.4 |
| ISO-EA | 0.68 | 0.53 | 0.47 | 1.00 | 0.96 | 0.52 | 0.92 | 0.53 | 0.70 | −1.4 |
| LE-EA | 0.68 | 0.54 | 0.47 | 1.00 | 0.96 | 0.53 | 0.92 | 0.51 | 0.70 | −1.4 |
| EBC-LE-EA | 0.68 | 0.55 | 0.47 | 0.95 | 0.96 | 0.52 | 0.93 | 0.53 | 0.70 | −1.4 |
| ncISO-EA | 0.68 | 0.53 | 0.47 | 1.00 | 0.96 | 0.47 | 0.92 | 0.53 | 0.69 | −2.8 |
| EBC-ISO-EA | 0.55 | 0.55 | 0.47 | 1.00 | 1.00 | 0.52 | 0.92 | 0.53 | 0.69 | −2.8 |
| MCE-EA | 0.68 | 0.54 | 0.47 | 0.95 | 0.93 | 0.51 | 0.92 | 0.52 | 0.69 | −2.8 |
| RA-ncISO-EA | 0.55 | 0.55 | 0.47 | 1.00 | 1.00 | 0.52 | 0.92 | 0.52 | 0.69 | −2.8 |
| RA-ISO-EA | 0.58 | 0.55 | 0.47 | 1.00 | 0.96 | 0.52 | 0.92 | 0.52 | 0.69 | −2.8 |
| RA-ncMCE-EA | 0.47 | 0.55 | 0.53 | 1.00 | 1.00 | 0.52 | 0.92 | 0.50 | 0.69 | −2.8 |
| LPCS | 0.55 | 0.55 | 0.53 | 1.00 | 0.96 | 0.52 | 0.93 | 0.51 | 0.69 | −2.8 |
| RA-LE-EA | 0.55 | 0.55 | 0.47 | 1.00 | 0.93 | 0.52 | 0.92 | 0.52 | 0.68 | −4.2 |
| RA-MCE-EA | 0.47 | 0.55 | 0.53 | 1.00 | 0.92 | 0.52 | 0.92 | 0.51 | 0.68 | −4.2 |
| HyperMap-CN | 0.52 | 0.55 | 0.41 | 1.00 | 0.86 | 0.57 | 0.89 | 0.46 | 0.66 | −7.0 |
| HyperMap | 0.52 | 0.60 | 0.32 | 1.00 | 0.92 | 0.49 | 0.90 | 0.46 | 0.65 | −8.5 |

Normalized mutual information (NMI) computed between the ground truth communities and the ones detected by the Infomap algorithm for eight real networks. NMI = 1 indicates a perfect match between the two partitions of the nodes. For each method, the network has been embedded in the hyperbolic space and the hyperbolic distances between the nodes are used to weight the observed links in the input matrix for the Infomap algorithm (see "Methods" section for details). As a reference, the Infomap algorithm has been run giving in input also the unweighted adjacency matrix, the related row is highlighted in italic. The table contains also some statistics for each network: number of nodes $N$, number of edges $E$, temperature $T$ (inversely related to the clustering coefficient), power-law degree distribution exponent $\gamma$, half of average degree $m$, and number of ground truth communities $N_c$. Due to the higher performance, only the EA methods are here reported, whereas the complete table is shown as Supplementary Table 2. The rightmost column reports the percentage of improvement with respect to the unweighted variant

embedding), and few algorithms can achieve these results in general[39]. On the other hand, significant Louvain improvements are obtained for the majority by the MCE-based approaches (Table 1): in Fig. 7b–d, we offer some real network examples of the embedding efficacy of these techniques for disclosing and visualizing the communities directly in the hyperbolic space. We gained perfect community detection also for another type of social network of larger size (composed of four hidden communities), the Opsahl_11[40] (Fig. 7c). This is a type of intra-organizational network where a link indicates that the connected employees have both awareness of each other's knowledge and skills on the job. The four hidden communities are related with the location of the employers (Paris, Frankfurt, Warsaw, and Geneva) and they were perfectly detected starting from the social-based network topology (Fig. 7c). For the Label propagation and Walktrap algorithms, the presence of a performance improvement given by some coalescent embedding techniques is confirmed (Supplementary Tables 3 and 4) and most of them are again ncISO-based and MCE-based approaches. Further discussions on the impact of the equidistant adjustment and on the results for large-size real networks are provided in Supplementary Discussion and Supplementary Tables 6–8.

**Beyond the two-dimensional space.** In comparison to the other approaches for hyperbolic embedding developed in previous studies and tailored for the two-dimensional hyperbolic disk, a peculiar property characterizes the class of unsupervised topological-based nonlinear machine learning algorithms adopted here. Being based on matrix decomposition methods for dimensionality reduction, there are not constraints on the number of dimensions that can be used to perform the embedding. This led us to investigate the possibility to enlarge the geometrical space from the hyperbolic disk to the hyperbolic sphere, with the addition of a further dimension.

Therefore, we have adopted the manifold-based unsupervised machine learning algorithms (LE, ISO, and ncISO) in order to extend the coalescent embedding to the three-dimensional hyperbolic space. After the pre-weighting step, the nonlinear dimension reduction is performed using an additional dimension with respect to the two-dimensional case. Considering a spherical coordinate system, the polar and azimuthal angles of the nodes in the hyperbolic sphere are preserved from the dimensionally reduced coordinates, whereas the radial coordinate is assigned as for the hyperbolic disk. The minimum-curvilinearity-based algorithms as well as the equidistant adjustment are not suitable for this extension, detailed explanations are provided in Supplementary Discussion. The analysis of the 3D hyperbolic embedding of the PSO networks highlighted the presence of common patterns, for which Fig. 8 and Supplementary Fig. 21 shows an explanatory example. At low temperature ($T = 0$), the nodes appear distributed over a well-defined closed 3D curve. Intuitively, it seems that the 2D hyperbolic disk already offers a perfect discrimination of the similarities and with the addition of the third dimension there is not much gain of information. With the increase of the temperature, the nodes look more and more spread around the closed 3D curve that was well defined at low temperature. Even if one angular dimension of the sphere still recovers most of the similarities present in the original network, it is unknown if the higher spread along the second angular

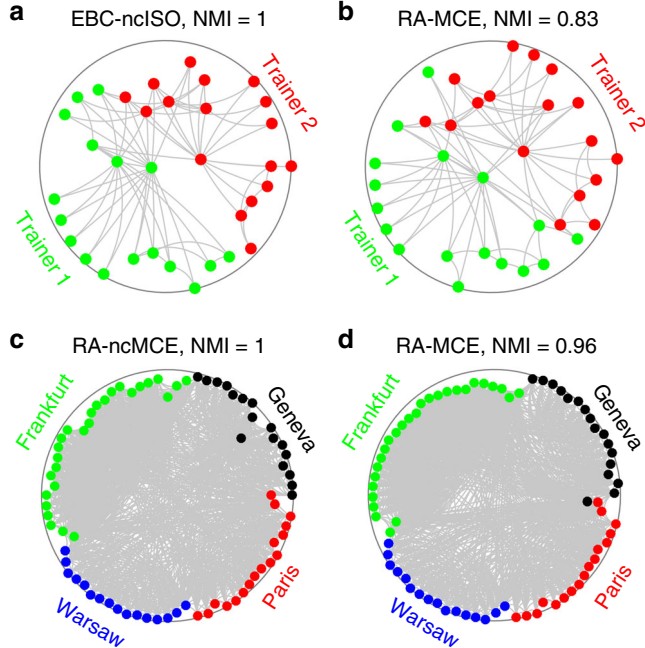

**Fig. 7** Communities in Karate and Opsahl_11 networks. **a**, **b** Karate network embedded with EBC-ncISO-EA and RA-MCE-EA. The network represents the friendship between the members of a university karate club in the United States. The two real communities are highlighted, they are formed by a split of the club into two parts, each following one trainer. The NMI obtained by the Louvain community detection algorithm is reported, where the embedding coordinates were used to weight the input matrix: observed links are weighted using the hyperbolic distances between the nodes and non-observed links using the hyperbolic shortest paths (see "Methods" section for details). NMI is the normalized mutual information and represents the shared information between two distributions, normalized between 0 and 1, where 1 indicates that the communities detected by the algorithm perfectly correspond to the ground truth communities (see "Methods" section for details). **c**, **d** Opsahl_11 network embedded with RA-ncMCE-EA and RA-MCE-EA. This is a type of intra-organizational network, where a link indicates that the connected employees have both awareness of each other's knowledge and skills on the job. The four real communities are highlighted, they are related with the location of the employers. All the approaches here adopted are adjusted according to EA strategy, although this is not explicitly reported in the subtitles for brevity. Note that the angular coordinates of the embedding in **b**, **d** have been aligned for a better visualization, respectively, to the ones in **a**, **c**, as described for the scatter plots in Fig. 4

dimension consists in a more refined discrimination between similar nodes or in noise. Since the original coordinates are 2D, this cannot be easily assessed.

In order to analyze the quality of the mapping from a quantitative point of view, the improvement given by the 3D hyperbolic embedding with respect to the 2D embedding is evaluated for the greedy routing and the community detection applications, the results are shown in Supplementary Tables 9–12 and commented in details in Supplementary Discussion.

Overall, the tests both on real and artificial networks represent a quantitative evidence that the addition of the third dimension of embedding in the hyperbolic space does not lead to a clear and significant improvement in performance. Although for the PSO model, this is indeed expected (because the synthetic networks are generated by a 2D geometrical model), we obtain the same result also on real networks, for which the hidden geometry is not

necessarily 2D. Therefore, we might conclude that in practical applications, at least on the tested networks, the 2D space appears to be enough for explaining the hidden geometry behind the complex network topologies. However, further investigations should be provided on networks of larger size and different types of origin, because the 3D space might conceptually offer an advantage with networks of large size. An additional interesting test can be to generate synthetic networks using a 3D PSO model, and then to compare the embedding accuracy using mapping techniques in 2D and 3D. Finally, we want to emphasize that, while the other hyperbolic embedding methods should be redesigned to fit for the 3D space, with the adoption of coalescent embedding approaches the exploration of additional dimensions of embedding is free of charge.

## Discussion

The investigation of the hidden geometry behind complex network topologies is an active research topic in recent years and the PSO model highlighted that the hyperbolic space can offer an adequate representation of the latent geometry of many real networks in a low dimensional space. However, in absence of a method able to map the network with high precision and in a reasonable computational time, any further analysis in the geometrical space would be compromised. Here we propose coalescent embedding: a class of unsupervised topological-based nonlinear machine learning algorithms for dimension reduction that offer a fast and accurate embedding in the hyperbolic space even for large graphs, this is the main product of the work. The embedding methods can be at the basis of any kind of investigation about network geometry and, as examples of applications, we presented community detection and greedy routing. However, the impact of coalescent embedding can be of importance for many disciplines including biology, medicine, computer science, and physics.

Below, we will summarize the main findings of this study. The first is that coalescent embedding significantly outperforms existing state-of-the-art methods for accuracy of mapping in hyperbolic space and, at the same time, reduces the computational complexity from $O(N^3)$–$O(N^4)$ of current techniques to $O(N^2)$. In addition, the results obtained on synthetic networks are indicative but should be considered with caution. In fact, LE-based coalescent embedding that performs better on synthetic networks is clearly outperformed in real network applications by MCE-based coalescent embedding. This implies that real networks might have a geometry that is even more tree like and hyperbolic (for this reason, MCE-based techniques can perform better on real networks) than the one hypothesized by the PSO model with uniform probability distribution of angular coordinates. In addition, although the topology of many real networks is certainly conditioned by the hyperbolic geometry, this is, however, one of the factors that shape their structure. Interestingly, good results are achieved also for networks with out of range $\gamma$ values. Since it has been demonstrated that a scale-free degree distribution is a necessary condition for hyperbolic geometry[41], this result demonstrates that the coalescent embedding methods can reach good performances also for networks whose latent geometry might be weakly hyperbolic or not hyperbolic.

The second important result is that the greedy routing performance on real networks embedded in the hyperbolic space using RA-ncMCE (which is a special type of coalescent embedding based on minimum curvilinearity) is only slightly inferior and in general comparable (no significant statistical difference is detected) to the one of networks mapped using the state-of-the-art methods. This is a remarkable result because the state-of-the-art methods directly optimize the hyperbolic distances in order to

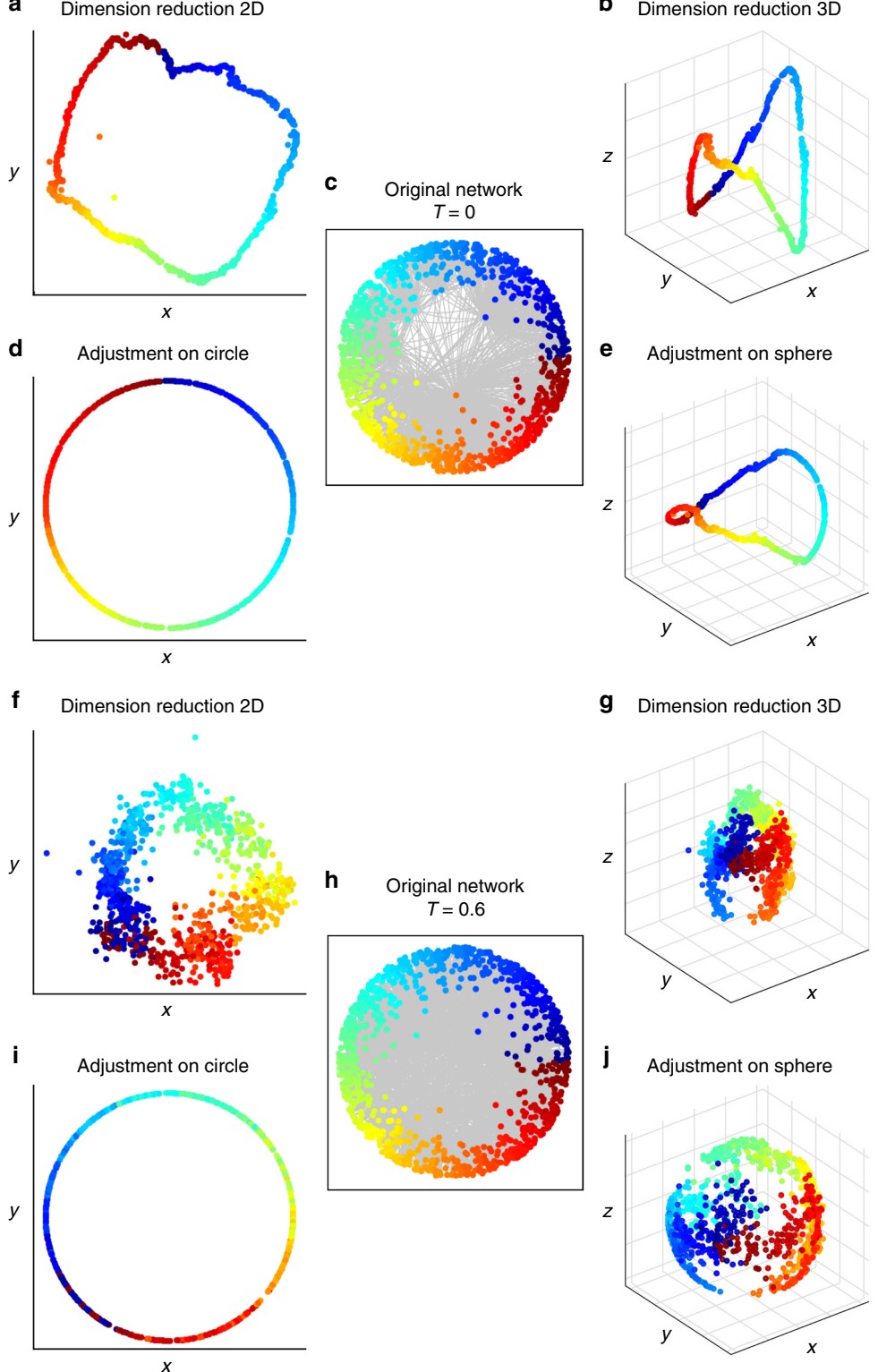

**Fig. 8** Comparison of 2D and 3D RA-ISO embedding for increasing temperature. The figure shows how the similarities of the original PSO network ($N = 1000$, $m = 6$, $\gamma = 2.5$) (**c**, **h**) are recovered either embedding in 2D (**a**, **f**) and arranging the angular coordinates over the circumference of a circle (**d**, **i**) or embedding in 3D (**b**, **g**) and adjusting the angular coordinates over a sphere (**e**, **j**). **a–e** At low temperature ($T = 0$), the nodes appear distributed over a well-defined closed 3D curve. Intuitively, it seems that the 2D hyperbolic disk already offers a perfect discrimination of the similarities and with the addition of the third dimension there is not much gain of information. **f–j** With the increase of the temperature ($T = 0.6$ reported, $T = 0.3$ and $T = 0.9$ shown in Supplementary Fig. 21), the nodes look more and more spread around the closed 3D curve that was well defined at low temperature. Even if one angular dimension of the sphere still recovers most of the similarities present in the original network, it is unknown if the higher spread along the second angular dimension consists in a more refined discrimination between similar nodes or in noise, therefore we have evaluated the improvement given by the 3D mapping for the greedy routing and the community detection applications

map connected nodes close and disconnected nodes far, which is a key factor for effective greedy routing. RA-ncMCE, instead, offers a comparable performance by ordering only the angular coordinates, which is a promising starting point to develop more effective strategies. In addition, RA-ncMCE can provide embedding of networks with 30,000 nodes in few minutes (Supplementary Fig. 22), hence can be of crucial aid for the investigation of greedy routing in real large networks. However, this last claim needs further analysis, because using the state-of-the-art methods on networks of 30,000 nodes was not computationally achievable in this study, therefore we cannot ensure that also on large networks RA-ncMCE provides a performance comparable to state-of-the-art methods such as HyperMap.

The third important finding of this study is that coalescent embedding can improve the performance of several state-of-the-art community detection algorithms when they are applied to real networks of small and medium size. But this improvement tends to be of less entity for large networks.

The previous state-of-the-art methods, such as HyperMap, were tailored for mapping networks in the hyperbolic space. The fourth key achievement of this study is that coalescent embedding techniques, although here adopted and tested for the inference of hyperbolic angular coordinates, are topological-based machine learning algorithms for nonlinear dimensionality reduction and therefore theoretically able to unfold any network latent geometry, not necessarily hyperbolic. The main point consists in the ability to design or learn a kernel that approximates the geometry of the hidden manifold behind the network topology. To this aim, we proposed two network pre-weighting node similarity methods (RA that is local topology based and EBC that is global topology based) aimed to approximate link geometrical information starting from the mere network topology. However, at the moment, we are making the first steps toward understanding these mechanisms of self-organization in complex networks, only further scientific efforts will help in the years to come to take advantage of the promising solutions proposed in this article to address many other questions in network geometry. For instance, the hidden geometry of networks with strong clustering and a non-scale-free degree distribution has been demonstrated to be Euclidean[23]. Thus, for networks with these characteristics, the Euclidean embedding obtained by the dimension reduction could be in theory directly adopted rather than exploiting it for the inference of the hyperbolic angular coordinates. However, this is just a speculation that needs to be proved in future studies. The fifth and last significant discovery of this study consists in an innovation introduced with coalescent embedding techniques that, although adopted for the mapping on a two-dimensional hyperbolic space, offers the possibility to explore spaces of higher dimensionality. Our simulations showed that the coalescent embedding on a three-dimensional hyperbolic sphere does not lead to a significant improvement on the tested data sets for tasks such as greedy routing and community detection. However, this result does not prevent further investigations that can lead to different results when the method is employed for embedding in arbitrary dimensions on new data and with different aims. Although in this study, only unweighted networks have been used, we let notice that the coalescent embedding methods can work also on weighted networks, where the weights suggest geometrical distances between the connected nodes, and this is a further advantage with respect to state-of-the-art methods such as HyperMap.

We want to stress that the exploitation of these machine learning techniques in complex network latent geometry is, as a matter of fact, initiated in this article, hence we suggest to take all the results here presented with a "grain of salt." Nevertheless, gathered all together, our findings suggest that the idea to connect unsupervised topological-based nonlinear machine learning theory with network latent geometry is a promising direction of research. In fact, coalescent embedding algorithms combine important performance improvement with a spectacular speed up both on in silico and real tests. We hope that this article will contribute to establish a new bridge at the interface between physics of complex networks and computational machine learning theory, and that future extended studies will dig into real applications revealing the impact of coalescent network embedding in social, natural, and life sciences. Interestingly, a first important instance of impact on network medicine is given in the article of Cacciola et al.[42], in which coalescent embedding in the hyperbolic space enhances our understanding of the hidden geometry of brain connectomes, introducing the idea of network latent geometry marker characterization of brain diseases with application in de novo drug naive Parkinson's disease patients.

## Methods

**Coalescent embedding algorithm in 2D**. Input: adjacency matrix, $x$
Output: polar coordinates of nodes, $(r, \theta)$

1. Pre-weighting rules
    1.1. Repulsion–attraction rule (RA, local)

$$x_{ij}^{\mathrm{RA1}} = \frac{d_i + d_j + d_i d_j}{1 + CN_{ij}} \quad (\text{if } x_{ij} = 1) \tag{1}$$

$$x_{ij}^{\mathrm{RA2}} = \frac{1 + e_i + e_j + e_i e_j}{1 + CN_{ij}} \quad (\text{if } x_{ij} = 1) \tag{2}$$

$x_{ij}$ value of $(i, j)$ entry in matrix $x$; $d_i$ degree of node $i$; $e_i$ external degree of node $i$ (links neither to $CN_{ij}$ nor to $j$); $CN_{ij}$ common neighbors of nodes $i$ and $j$.

 1.2. Edge betweenness centrality rule (EBC, global)

$$x_{ij}^{\mathrm{EBC}} = \sum_{s,t \epsilon V} \frac{\sigma(s, t | l_{ij})}{\sigma(s, t)} \tag{3}$$

$V$ set of nodes; $s$, $t$ any combination of network nodes in $V$; $\sigma(s, t)$ number of shortest paths $(s, t)$; $\sigma(s, t | l_{ij})$ number of shortest paths $(s, t)$ through link $l_{ij}$.

2. Nonlinear dimension reduction
    2.1 Laplacian eigenmaps (LE), 2nd–3rd dimensions
    2.2 Isomap (ISO), 1st–2nd dimensions
    2.3 Noncentered Isomap (ncISO), 2nd–3rd dimensions
    2.4 Minimum curvilinearity (MCE), 1st dimension
    2.5 Noncentered minimum curvilinearity (ncMCE), 2nd dimension
3. Angular coordinates ($\theta$)
    3.1 Circular adjustment
    3.2 Equidistant adjustment
4. Radial coordinates ($r$)

Nodes are sorted according to descending degree and the radial coordinate of the $i$-th node in the set is computed according to:

$$r_i = \frac{2}{\zeta}[\beta \ln i + (1 - \beta) \ln N] \quad i = 1, 2, \dots, N \tag{4}$$

$N$ number of nodes; $\zeta = \sqrt{-K}$, we set $\zeta = 1$; $K$ curvature of the hyperbolic space; $\beta = \frac{1}{\gamma - 1}$ popularity fading parameter; $\gamma$ exponent of power-law degree distribution.
A flow chart with the visualization of the intermediate results produced by the algorithmic steps on a toy network are provided in Fig. 2.

**Coalescent embedding algorithm in 3D**. Input: adjacency matrix, $x$
Output: spherical coordinates of nodes, $(r, \theta, \phi)$

1. Pre-weighting rules (same as 2D)
    1.1. Repulsion–attraction rule (RA, local)
    1.2. Edge betweenness centrality rule (EBC, global)
2. Nonlinear dimension reduction
    2.1 Laplacian eigenmaps (LE), 2nd-3rd-4th dimensions
    2.2 Isomap (ISO), 1st-2nd-3rd dimensions
    2.3 Noncentered Isomap (ncISO), 2nd-3rd-4th dimensions
3. Angular coordinates ($\theta$, $\phi$)
    The angular coordinates obtained from the nonlinear dimension reduction are used.
4. Radial coordinates ($r$) (same as 2D)
    Notes:

Four variants of the RA pre-weighting rule have been tested, the results are shown in Supplementary Fig. 23 and commented in Supplementary Discussion. Here only the two best rules are reported.

The exponent $\gamma$ of the power-law degree distribution has been fitted using the MATLAB script "plfit.m." according to a procedure described by Clauset et al.[43] and released at http://www.santafe.edu/~aaronc/powerlaws/.

**Manifold-based embedding**. The first type of topological-based unsupervised machine learning for nonlinear dimension reduction adopted in this study are Isomap, ISO[27], and Laplacian eigenmaps LE[28]. These two methods are manifold-based machine learning because, in classical dimension reduction of multi-dimensional data sets, they approximate the sample data manifold using a proximity graph, and then they embed by matrix decomposition the sample distances in a 2D space. In our application the proximity graph is already given, representing an important advantage, because the topological connections (similarities) between the nodes are already known. In fact, the problem to infer a proximity graph is not trivial and generally requires the introduction of at least a tuning parameter, for instance, in the procedure to learn a nearest-neighbor graph (network) that approximates the manifold. Furthermore, there is not a clear strategy to unsu-pervisedly tune these kinds of parameters to infer the proximity graph.

ISO is based on extracting a distance matrix (or kernel) that stores all the network shortest path distances (also named geodesic distances) that approximate the real distances over the manifold. Then the kernel is centered and in this work singular value decomposition (SVD) is applied to embed the nodes in the 2D space. We also propose the noncentered version of the same algorithm, named ncISO, in which the kernel centering is neglected. Consequently, the first dimension of embedding is discarded because, since it points toward the center of the manifold, is not useful. For more computational details on the implementation of ISO, please refer to refs. [8, 26, 27].

LE is a different type of manifold machine learning. In fact, the inference of a distance kernel (for instance, the shortest path kernel for ISO) starting from the network structure is not required in this algorithm, which makes it faster than ISO. Indeed, the idea behind LE is to perform the eigen-decomposition of the network's Laplacian matrix, and then to perform 2D embedding of the network's nodes according to the eigenvectors related to the second and third smallest eigenvalues. The first smallest eigenvalue is zero, thus the related eigenvector is neglected. In order to implement a weighted version of this algorithm we used, as suggested in the original publication[28], the "heat-function" (instead of the pre-weighting values as they are in their original scale):

$$\tilde{x}_{ij} = e^{-\frac{x_{ij}^2}{t}} \qquad (5)$$

Where $x_{ij}$ is the original pre-weighing value for the link $i$, $j$, and $t$ is a scaling factor fixed as the squared mean of all the network's pre-weighting values. For more computational details on the implementation of LE, please refer to ref. [28].

**Minimum curvilinearity and minimum curvilinear embedding**. The centered and noncentered versions of the minimum curvilinear embedding algorithm—respectively named MCE and ncMCE—are based on a general nonlinear dissimilarity learning theory called minimum curvilinearity[26]. These approaches compress the information in one unique dimension because they learn nonlinear similarities by means of the minimum spanning tree, providing a hierarchical-based mapping. This is fundamentally different from the previous algorithms (Isomap and LE), which are manifold-based. If we would consider the mere unsupervised machine learning standpoint, we would notice that manifold-based techniques in this study showed two main weaknesses: (1) they offer less compression power because two orthogonal dimensions of representation, instead of one, are needed; (2) the node similarity pattern remains nonlinear (circular) also in the embedded space, thus the goal of the nonlinear dimension reduction to linearize a (hidden) nonlinear pattern in the embedding space is missed.

In unsupervised tasks where the objective is to discover unknown and unexpected sample stratifications—for instance, the discovery of unforeseen groups of patients with undetected molecular-based disease variations—the linearization of a pattern along one unique embedding dimension can offer an undisputed help to recognize hidden and hierarchical organized subgroups[26, 44].

However, the utility of a computational technique varies in relation to the designed target to reach. In fact, when the goal is to perform embedding of the points of a multidimensional data set for unsupervised pattern detection and linearization, the graph that connects the points and approximates the manifold is unknown, and the minimum-curvilinearity-based techniques offer an advantage over manifold approaches. Conversely, when the task is to embed a network in the hyperbolic space, the graph that approximates the manifold is already given, and the goal of embedding is to retain—and not to linearize—the circular node similarity pattern in the 2D space. Therefore, the manifold-based techniques, compared to the minimum-curvilinearity-based, could offer a better approximation of the angular coordinates (representing the network node similarities) especially for high temperatures of the PSO model, when the tree-like network organization and the related hyperbolic geometry degenerate.

Interestingly, MCE and ncMCE were theoretically designed by Cannistraci et al.[8, 26] according to a previous theory of Bogugna et al. presented in the

"Navigability of complex networks"[2]. This article was clearly explaining that to navigate efficiently a network (and thus approximate geodesic/curvilinear pairwise node connections over the hidden manifold) it was not necessary to know the complete information of network topology in the starting point of the navigation. A greedy routing process (thus, based on the neighborhood information) was enough to efficiently navigate the network. This triggered an easy conclusion at the basis of the minimum curvilinearity learning theory design: to approximate curvilinear distances between the points of the manifold it is not necessary to reconstruct the nearest-neighbor graph. Just a greedy routing process (that exploits a norm, for instance, Euclidean) between the points in the multidimensional space, is enough to efficiently navigate the hidden network that approximates the manifold in the multidimensional space. In few words, learning nonlinear distances over the manifold by navigating an invisible and unknown network was possible, because the navigation process was instead guided by a greedy routing. Hence, according to Cannistraci et al.[8, 26], a preferable greedy routing strategy was the minimum spanning tree (MST). The only hypothesis of application of this approach was that the points were not homogenously distributed in a lattice regular structure, or a similar degenerative condition. Thus, the minimum curvilinear kernel is the matrix that collects all the pairwise distances between the points (or nodes) computed over the MST. And the ncMCE is the embedding of the noncentered minimum curvilinear kernel by means of the SVD. The reason why to exploit the ncMCE—noncentered version of MCE[26]—is discussed in a second article[8] that presents how to use this approach for link prediction in protein interaction networks. The main difference between MCE and ncMCE is that in general MCE linearizes the hidden patterns along the first dimension of embedding while ncMCE along the second dimension (since it is noncentered, the first dimension of embedding should be generally neglected because it points toward the center of the manifold). To conclude this part, MCE/ncMCE are conceptually different from all the other approaches because they are one of the few (maybe the only, to the best of our knowledge) dimensionality reduction methods that performs hierarchical embedding, and they exploit the MST as a highway to navigate different regions of the network. Although they exploit a small fraction of the network links—practically only the MST, which consists of N-1 links in a network with N nodes—the reason why they work efficiently to infer the angular coordinates of networks that follow the PSO model is well explained in the article of Papadopoulos et al.[29]; thus, we take the advantage to report the full paragraph: <<This work shows that random geometric graphs[45] in hyperbolic spaces are an adequate model for complex networks. The high-level explanation of this connection is that complex networks exhibit hierarchical, tree-like organization, while hyperbolic geometry is the geometry of trees[46]. Graphs representing complex networks appear then as discrete samples from the continuous world of hyperbolic geometry.>>

However, the problem to compute the MST on an unweighted adjacency matrix is that we do not have a norm that suggests the hidden connectivity geometry. Thus, there was a clear margin to improve the performance of MCE/ncMCE by pre-weighting the links in the network (and the adjacency matrix) using a convenient strategy to suggest topological distances between the connected nodes. In fact, in the Supplementary Figs. 5 and 6, we notice that the pre-weighting strategy significantly boosts MCE/ncMCE performance.

**HyperMap**. HyperMap[29] is a method to map a network into its hyperbolic space based on maximum likelihood estimation (MLE). For sake of clarity, the first algorithm for MLE-based network embedding in the hyperbolic space is not HyperMap, but to the best of our knowledge is the algorithm proposed by Boguñá et al. in ref. [12]. HyperMap is basically an extension of that method applied to the PSO model. Unlike the coalescent embedding techniques, it can only perform the embedding in two dimensions and cannot exploit the information of the weights. It replays the hyperbolic growth of the network and at each step it finds the polar coordinates of the added node by maximizing the likelihood that the network was produced by the E-PSO model[29].

For curvature $K = -1$, the procedure is as follows:

(1) Nodes are sorted decreasingly by degree and then labeled $i = 1, 2, \ldots, N$ according to the order;

(2) Node $i = 1$ is born and assigned radial coordinate $r_1 = 0$ and a random angular coordinate $\theta_1 \in [0, 2\pi]$;

(3) For each node $i = 2, 3, \ldots, N$ do:

(3.a) Node $i$ is added to the network and assigned a radial coordinate

$$r_i = 2 \ln i \qquad (6)$$

(3.b) The radial coordinate of every existing node $j < i$ is increased according to

$$r_j(i) = \beta r_j + (1 - \beta) r_i \qquad (7)$$

where $\beta \in (0, 1]$ is obtained from the exponent $\gamma$ of the power-law degree

distribution

$$\gamma = 1 + \frac{1}{\beta} \qquad (8)$$

(3.c) The node $i$ is assigned an angular coordinate by maximizing the likelihood

$$L_i = \prod_{1 \le j < i} p(h_{ij})^{x_{ij}} \left(1 - p(h_{ij})\right)^{1 - x_{ij}} \qquad (9)$$

where $p(h_{ij})$ is the connection probability of nodes $i$ and $j$

$$p(h_{ij}) = \frac{1}{1 + \exp\left(\frac{h_{ij} - R_i}{2T}\right)} \qquad (10)$$

which is function of the hyperbolic distance $h_{ij}$ between node $i$ and node $j$, the current radius of the hyperbolic disk $R_i$, and the network temperature $T$[29]. The maximization is done by numerically trying different angular coordinates in steps of $2\pi/N$ and choosing the one that leads to the biggest $L_i$. The method has been implemented in MATLAB.

**HyperMap-CN**. HyperMap-CN[30] is a further development of HyperMap, where the inference of the angular coordinates is not performed anymore maximizing the likelihood $L_{i,L}$, based on the connections and disconnections of the nodes, but using another local likelihood $L_{i,CN}$, based on the number of common neighbors between each node $i$ and the previous nodes $j < i$ at final time. Here the hybrid model has been used, a variant of the method in which the likelihood $L_{i,CN}$ is only adopted for the high-degree nodes and $L_{i,L}$ for the others, yielding a shorter running time. Furthermore, a speed-up heuristic and corrections steps can be applied. The speed up can be achieved by getting an initial estimate of the angular coordinate of a node $i$ only considering the previous nodes $j < i$ that are $i$'s neighbors. The maximum likelihood estimation is then performed only looking at an interval around this initial estimate. Correction steps can be used at predefined times $i$ after step 3.c (in the description of HyperMap). Each existing node $j < i$ is visited and with the knowledge of the rest of the coordinates, the angle of $j$ is updated to the value that maximizes the likelihood $L_{j,L}$. The C++ implementation of the method has been released by the authors at the website https://bitbucket.org/dk-lab/2015_code_hypermap. For the embedding of all the PSO networks, the default settings (correction steps but no speed-up heuristic) have been used, whereas for all the real networks neither correction steps nor speed-up heuristic have been used.

**LPCS**. Link prediction with community structure (LPCS)[31] is a hyperbolic embedding technique that consists of the following steps: (1) Detect the hierarchical organization of communities. (2) Order the top-level communities starting from the one that has the largest number of nodes and using the community intimacy index, which takes into account the proportion of edges within and between communities. (3) Recursively order the lower-level communities based on the order of the higher-level communities, until reaching the bottom level in the hierarchy. (4) Assign to every bottom-level community an angular range of size proportional to the nodes in the community, in order to cover the complete circle with non-overlapping angular ranges. Sample the angular coordinates of the nodes uniformly at random within the angular range of the related bottom-level community. (5) Assign the radial coordinates according to Eq. (4).

The LPCS code first takes advantage of the Louvain R function for detecting the hierarchy of communities (see Louvain method), then we implemented the embedding in MATLAB.

**The C-score for angular coordinates evaluation**. The inference of the angular coordinates order is evaluated according to a modified version of the concordance score (C-score)[47]. The C-score is defined as the proportion of sample pairs for which the ranking by a prediction model corresponds to the true ranking. Here, we need to adapt the interpretation to our particular case, where the ranked samples are not disposed along an axis but on a circle, hence the C-score can be interpreted as the proportion of node pairs for which the angular relationship in the inferred network corresponds to the angular relationship in the original network. Below, we report the formula to compute the C-score in our case:

$$C-score = \frac{\sum_{i=1}^{n-1} \sum_{j=i+1}^{n} \delta(i,j)}{n \times (n-1)/2} \qquad (11)$$

Where: $n$ is the total number of network nodes; $i$ and $j$ indicate two nodes and $\delta(i, j)$ is 1 if the shortest angular distance from $i$ to $j$ has the same direction (clockwise or counter-clockwise) in both the original and inferred network, and 0 if the direction is the opposite in the original and inferred coordinates.

Since in the inferred network, the nodes could have been arranged in the opposite clock direction with respect to the original network, the C-score is computed also considering the inferred angular relationships in the opposite clock direction (the two conditions for the value of $\delta(i, j)$ are inverted) and the maximum between the two values is chosen.

**Greedy routing**. An important characteristic that can be studied in a network embedded in a geometrical space is its navigability. The network is considered navigable if the greedy routing (GR) performed using the node coordinates in the geometrical space is efficient[2]. In the GR, for each pair of nodes $i$ and $j$, a packet is sent from $i$ with destination $j$. Each node knows only the address of its neighbors and the address of the destination $j$, which is written in the packet. The address of a node is represented by its coordinates in the geometrical space. In the GR procedure adopted[29], the nodes are located in the hyperbolic disk and at each hop the packet is forwarded from the current node to its neighbor closest to the destination, meaning at the lowest hyperbolic distance. The packet is dropped when this neighbor is the same from which the packet has been received at the previous hop, since a loop has been generated. In order to evaluate the efficiency of the GR, two metrics are usually taken into account: the percentage of successful paths and the average hop-length of the successful paths[2]. The first one indicates the proportion of packets that are able to reach their destinations—the higher the better—whereas the second one indicates if the successful packets require on average a short path to reach the destination—the lower the better. In order to compare the performance of methods in a unique way while taking both of the metrics into account, a GR-score has been introduced and it is computed as follows:

$$GR_{score} = \frac{\sum_{i=1}^{n} \sum_{j=1, j \ne i}^{n} \frac{sp_{ij}}{p_{ij}}}{n \times (n-1)} \qquad (12)$$

Where $i$ and $j$ are two within the set of $n$ nodes, $sp_{ij}$ is the shortest path length from $i$ to $j$ and $p_{ij}$ is the GR path length from $i$ to $j$. The ratio $\frac{sp_{ij}}{p_{ij}}$ assumes values in the interval [0, 1]. When the GR is unsuccessful, the path length is infinite and therefore the ratio is 0, which represents the worst case. When the GR is successful, the path length is greater than 0 and tends to 1 as the path length tends to the shortest path length, becoming 1 in the best case. The GR-score is the average of this ratio over all the node pairs.

**Louvain algorithm for community detection**. The Louvain algorithm[34] is separated into two phases, which are repeated iteratively.

At first, every node in the (weighted) network represents a community in itself. In the first phase, for each node $i$, it considers its neighbors $j$ and evaluates the gain in modularity that would take place by removing $i$ from its community and placing it in the community of $j$. The node $i$ is then placed in the community $j$ for which this gain is maximum, but only if the gain is positive. If no gain is possible, node $i$ stays in its original community. This process is applied until no further improvement can be achieved.

In the second phase, the algorithm builds a new network whose nodes are the communities found in the first phase, whereas the weights of the links between the new nodes are given by the sum of the weight of the links between nodes in the corresponding two communities. Links between nodes of the same community lead to self-loops for this community in the new network.

Once the new network has been built, the two phase process is iterated until there are no more changes and a maximum of modularity has been obtained. The number of iterations determines the height of the hierarchy of communities detected by the algorithm.

For each hierarchical level, there is a possible partition to compare to the ground truth annotation. In this case, the hierarchical level considered is the one that guarantees the best match, therefore the detected partition that gives the highest NMI value.

We used the R function "multilevel.community," an implementation of the method available in the "igraph" package[48].

In this study, the embedding of the network in the hyperbolic space has been exploited in order to weight the input adjacency matrix. Given the hyperbolic coordinates, the observed links have been weighted using the formula

$$x_{ij}^{HD} = \frac{1}{1 + HD_{ij}} \qquad (13)$$

where $HD_{ij}$ is the hyperbolic distance between nodes $i$ and $j$. For the Louvain algorithm, a further variant has been tested in which also the non-observed links have been weighted using the formula

$$x_{ij}^{HSP} = \frac{1}{1 + HSP_{ij}} \qquad (14)$$

where $HSP_{ij}$ is the hyperbolic shortest path between nodes $i$ and $j$, computed as the sum of the hyperbolic distances over the shortest path.

**Infomap algorithm for community detection**. The Infomap algorithm[35] finds the community structure by minimizing the expected description length of a random walker trajectory using the Huffman coding process[49].

It uses the hierarchical map equation (a further development of the map equation, to detect community structures on more than one level), which indicates the theoretical limit of how concisely a network path can be specified using a given partition structure. In order to calculate the optimal partition (community

structure, this limit can be computed for different partitions and the community annotation that gives the shortest path length is chosen.

For each hierarchical level, there is a possible partition to compare to the ground truth annotation. In this case, the hierarchical level considered is the one that guarantees the best match, therefore the detected partition that gives the highest NMI value.

We used the C implementation released by the authors at http://www.mapequation.org/code.html.

In this study, the embedding of the network in the hyperbolic space has been exploited in order to weight the input adjacency matrix. Given the hyperbolic coordinates, the observed links have been weighted using the hyperbolic distances according to Eq. (13).

**Label propagation algorithm for community detection**. The label propagation algorithm[36] initializes each node with a unique label and iteratively updates each node label with the one owned by the majority of the neighbors, with ties broken uniformly at random. The update is performed in an asynchronous way and the order of the nodes at each iteration is chosen randomly. As the labels propagate through the network, densely connected groups of nodes quickly reach a consensus on a unique label. The iterative process stops when every node has the same label as the majority its neighbors, ties included. At the end of the procedure, the nodes having the same label are grouped together to form a community. Since the aim is not the optimization of an objective function and the propagation process contains randomness, there are more possible partitions that satisfy the stop criterion and therefore the solution is not unique. For this reason, the algorithm has been run for 10 independent iterations and the mean performance is reported.

We used the R function "label.propagation.community," an implementation of the method available in the "igraph" package[48].

In this study, the embedding of the network in the hyperbolic space has been exploited in order to weight the input adjacency matrix. Given the hyperbolic coordinates, the observed links have been weighted using the hyperbolic distances according to Eq. (13).

**Walktrap algorithm for community detection**. The Walktrap algorithm[37] is based on an agglomerative method for hierarchical clustering: the nodes are iteratively grouped into communities exploiting the similarities between them. The nodes similarities are obtained using random walks and are based on the idea that random walks tend to get trapped into densely connected subgraphs corresponding to communities.

The agglomerative method uses heuristics to choose which communities to merge and implements an efficient way to update the distances between communities. At the end of the procedure, a hierarchy of communities is obtained and each level offers a possible partition. The algorithm chooses as final result the partition that maximizes the modularity.

We used the R function "walktrap.community," an implementation of the method available in the "igraph" package[48].

In this study, the embedding of the network in the hyperbolic space has been exploited in order to weight the input adjacency matrix. Given the hyperbolic coordinates, the observed links have been weighted using the hyperbolic distances according to Eq. (13).

**Normalized mutual information**. The evaluation of the community detection has been performed using the normalized mutual information (NMI) as in ref. [50]. The entropy can be defined as the information contained in a distribution $p(x)$ in the following way:

$$H(X) = \sum_{x \in X} p(x) \log p(x) \tag{15}$$

The mutual information is the shared information between two distributions:

$$I(X, Y) = \sum_{y \in Y} \sum_{x \in X} p(x, y) \log \left( \frac{p(x, y)}{p_1(x) p_2(y)} \right) \tag{16}$$

To normalize the value between 0 and 1 the following formula can be applied:

$$\mathrm{NMI} = \frac{I(X, Y)}{\sqrt{H(X) H(Y)}} \tag{17}$$

If we consider a partition of the nodes in communities as a distribution (probability of one node falling into one community), we can compute the matching between the annotation obtained by the community detection algorithm and the ground

truth communities of a network as follows:

$$H(C_D) = \sum_{h=1}^{n_D} \frac{n_h^D}{N} \log \left( \frac{n_h^D}{N} \right) \tag{18}$$

$$H(C_T) = \sum_{l=1}^{n_T} \frac{n_l^T}{N} \log \left( \frac{n_l^T}{N} \right) \tag{19}$$

$$I(C_D, C_T) = \sum_h \sum_l \frac{n_{h,l}}{N} \log \left( \frac{n_{h,l}}{n_h^D n_l^T} \right) \tag{20}$$

$$\mathrm{NMI}(C_D, C_T) = \frac{I(C_D, C_T)}{\sqrt{H(C_D) H(C_T)}} \tag{21}$$

Where:

$N$—number of nodes;

$n^D$, $n^T$—number of communities detected by the algorithm ($D$) or ground truth ($T$);

$n_{h,l}$—number of nodes assigned to the $h$-th community by the algorithm and to the $l$-th community according to the ground truth annotation.

We used the MATLAB implementation available at http://commdetect.weebly.com/. As suggested in the code, when $\frac{N}{n^T} \leq 100$, the NMI should be adjusted in order to correct for chance[51].

**Generation of synthetic networks using the PSO model**. The popularity-similarity-optimization (PSO) model[1] is a generative network model recently introduced to describe how random geometric graphs grow in the hyperbolic space. In this model, the networks evolve optimizing a trade-off between node popularity, abstracted by the radial coordinate, and similarity, represented by the angular coordinate distance, and they exhibit many common structural and dynamical characteristics of real networks.

The model has four input parameters:

$m > 0$, which is equal to half of the average node degree;

$\beta \in (0, 1]$, defining the exponent $\gamma = 1 + 1/\beta$ of the power-law degree distribution;

$T \geq 0$, which controls the network clustering; the network clustering is maximized at $T = 0$, it decreases almost linearly for $T = [0, 1)$ and it becomes asymptotically zero if $T > 1$;

$\zeta = \sqrt{-K} > 0$, where $K$ is the curvature of the hyperbolic plane. Since changing $\zeta$ rescales the node radial coordinates and this does not affect the topological properties of networks[1], we considered $K = -1$.

Building a network of $N$ nodes on the hyperbolic disk requires the following steps: (1) Initially the network is empty. (2) At time $i = 1, 2, ..., N$ a new node $i$ appears with radial coordinate as described in Eq. (6) and angular coordinate $\theta_i$ uniformly sampled in $[0, 2\pi]$; all the existing nodes $j < i$ increase their radial coordinates according to Eq. (7) in order to simulate popularity fading. (3) If $T = 0$, the new node connects to the $m$ hyperbolically closest nodes; if $T > 0$, the new node picks a randomly chosen existing node $j < i$ and, given that it is not already connected to it, it connects to it with probability $p(h_{ij})$ (see Eq. (10)), repeating the procedure until it becomes connected to $m$ nodes. (4) The growing process stops when $N$ nodes have been introduced.

**Real networks data set**. The community detection methods have been tested on eight small real networks, which represent differing systems, and on eight large Internet networks.

The networks have been transformed into undirected, unweighted, without self-loops and only the largest connected component has been considered. The information of their ground truth communities is available. Every table, together with the results, provides also some basic statistics of the networks.

The first small network is about the Zachary's Karate Club[38], it represents the friendship between the members of a university karate club in the United States. The communities are formed by a split of the club into two parts, each following one trainer.

The networks from the second to the fifth are intra-organizational networks from ref. [40] and can be downloaded at https://toreopsahl.com/datasets/#Cross_Parker. Opsahl_8 and Opsahl_9 come from a consulting company and nodes represent employees. In Opsahl_8, employees were asked to indicate how often they have turned to a co-worker for work-related information in the past, where the answers range from: 0—I do not know that person; 1—Never; 2—Seldom; 3—Sometimes; 4—Often; 5—Very often. Directions were ignored. The data were turned into an unweighted network by setting a link only between employees that have at least asked for information seldom (2).

In the Opsahl_9 network, the same employees were asked to indicate how valuable the information they gained from their co-worker was. They were asked to show how strongly they agree or disagree with the following statement: "In general, this person has expertise in areas that are important in the kind of work I do." The weights in this network are also based on the following scale: 0—Do not know this

person; 1—Strongly disagree; 2—Disagree; 3—Neutral; 4—Agree; 5—Strongly agree. We set a link if there was an agreement (4) or strong agreement (5). Directions were ignored.

The Opsahl_10 and Opsahl_11 networks come from the research team of a manufacturing company and nodes represent employees. The annotated communities indicate the company locations (Paris, Frankfurt, Warsaw, and Geneva).

For Opsahl_10, the researchers were asked to indicate the extent to which their co-workers provide them with information they use to accomplish their work. The answers were on the following scale: 0—I do not know this person/I never met this person; 1—Very infrequently; 2—Infrequently; 3—Somewhat frequently; 4—Frequently; 5—Very frequently. We set an undirected link when there was at least a weight of 4.

For Opsahl_11, the employees were asked about their awareness of each other's knowledge ("I understand this person's knowledge and skills. This does not necessarily mean that I have these skills and am knowledgeable in these domains, but I understand what skills this person has and domains they are knowledgeable in."). The weighting was on the scale: 0—I do not know this person/I have never met this person; 1—Strongly disagree; 2—Disagree; 3—Somewhat disagree; 4—Somewhat agree; 5—Agree; 6—Strongly agree. We set a link when there was at least a 4, ignoring directions.

The Polbooks network represents frequent co-purchases of books concerning US politics on amazon.com. Ground truth communities are given by the political orientation of the books as either conservative, neutral or liberal. The network is unpublished but can be downloaded at http://www-personal.umich.edu/~mejn/netdata/, as well as with the Karate, Football, and Polblogs networks.

The Football network[32] presents games between division IA colleges during regular season fall 2000. Ground truth communities are the conferences that each team belongs to.

The Polblogs[52] network consists of links between blogs about the politics in the 2004 US presidential election. The ground truth communities represent the political opinions of the blogs (right/conservative and left/liberal).

The large size networks considered for community detection are autonomous systems (AS) Internet topologies extracted from the data collected by the Archipelago active measurement infrastructure (ARK) developed by CAIDA[53]. The connections in the topology are not physical but logical, representing AS relationships, and the annotated communities are the geographical locations (countries). ARK200909-ARK201012 are topologies collected from September 2009 to December 2010 at time steps of 3 months (download available at https://bitbucket.org/dk-lab/2015_code_hypermap/src/bd473d7575c35e099b520bf669d92aea81fac69b/AS_topologies/). AS201501_IPv4 is a more recent version of the IPv4 Internet topology, collected on January 2015 (download at http://www.caida.org/data/active/ipv4_routed_topology_aslinks_dataset.xml). AS201501_IPv6 is as recent as the previous one but represents the IPv6 Internet network (download at https://www.caida.org/data/active/ipv6_allpref_topology_dataset.xml).

**Complexity of coalescent embedding algorithms**. The time complexity of the coalescent embedding algorithms proposed can be obtained summing up the computational cost of the main steps: pre-weighting, dimension reduction, assignment of angular coordinates, and radial coordinates. All the complexities reported assume the network to be connected, therefore the number of edges $E$ has at least the same order of complexity as the number of nodes $N$.

1. Pre-weighting rules
   RA: it only requires basic operations on sparse matrices, whose complexity is proportional to the number of nonzero elements, therefore $O(E)$.
   EBC: it requires the computation of the EBC, which takes $O(EN)$ using the Brandes' algorithm for unweighted graphs[54].
2. Dimension reduction techniques
   2.1) LE: the method performs the eigen-decomposition of the Laplacian matrix solving a generalized eigenvalue problem and then uses the eigenvectors related to the smallest eigenvalues, discarding the first because it is zero. The computation of all the eigenvalues of the matrix requires $O(N^3)$[55], using the MATLAB function "eig." However, since the Laplacian matrix is sparse and only a few eigenvalues are needed ($k = 3$ for a 2D embedding and $k = 4$ for a 3D embedding in the hyperbolic space), the MATLAB function "eigs" can be executed. First, the matrix is factorized, which requires $O(N^{3/2})$ for sparse matrices[56], then it uses the implicitly restarted Arnoldi method (IRAM)[57] as implemented in ARPACK[58]. It is an iterative method whose convergence speed strictly depends on the relative gap between the eigenvalues, which makes the computational complexity difficult to analyze in terms of $N$ and $E$.
   2.2) ISO: the method computes as first all the pairwise shortest paths between the nodes using the Johnson's algorithm, which takes $O(EN)$[59]. The obtained kernel is centered, which costs $O(N^2)$ due to operations on dense matrices, and finally singular value decomposition is applied, in order to obtain the singular vectors related to the largest singular values. The computation of all the singular values of the matrix requires $O(N^3)$[55], using the MATLAB function "svd." However, since only a few singular values are needed ($k = 2$ for a 2D embedding and $k = 3$ for a 3D embedding in the

hyperbolic space), the function "lansvd" from the software package PROPACK can be exploited, which has a lower computational complexity equal to $O(kN^2)$[60].
   2.3) ncISO: the method performs the same operations as ISO, omitting the kernel centering. As a consequence, the first dimension of embedding is discarded, therefore the singular values to compute are $k = 3$ for a 2D embedding and $k = 4$ for a 3D embedding in the hyperbolic space.
   2.4) MCE: the method computes as first a minimum spanning tree over the network using the Kruskal's algorithm, whose complexity is $O(E + X * \log (N))$, where $X$ is the number of edges no longer than the longest edge in the MST[61]. Starting from the minimum spanning tree, all the pairwise transversal distances between disconnected nodes are computed in order to form the MC-Kernel, the complexity of this step it $O(EN)$ with $E = (N - 1)$ hence $O(N^2)$. The following step is the kernel centering that costs $O(N^2)$. Since the angular coordinates are inferred according to the first dimension of embedding, the singular values to compute are $k = 1$ for a 2D embedding in the hyperbolic space.
   2.5) ncMCE: the method performs the same operations as MCE, omitting the kernel centering. As a consequence, the first dimension of embedding is discarded and the angular coordinates are inferred according to the second one. Therefore, the singular values to compute are $k = 2$ for a 2D embedding in the hyperbolic space.
3. Angular coordinates
   3.1) Circular adjustment: the assignment requires the conversion from Cartesian to polar coordinates for the methods LE, ISO, and ncISO and a rescaling of the coordinates for MCE and ncMCE, in both the cases the cost is $O(N)$.
   3.2) Equidistant adjustment: a sorting operation is performed, which costs $O(N \log N)$.
4. Radial coordinates: the assignment is performed applying a given mathematical formula, which requires the nodes to be sorted by degree, the cost is therefore $O(N \log N)$.

Summarizing, if full matrix factorization techniques are used, all the methods have a complexity of $O(N^3)$. Instead, if the truncated variants are used, for ISO, ncISO, MCE, and ncMCE, the SVD takes $O(N^2)$ since $k$ is a small constant, therefore they have a complexity dominated by the shortest path computation, which is $O(EN)$ in case of ISO and ncISO and $O(N^2)$ in case of MCE and ncMCE. The slowest pre-weighting is EBC and it is in the same order of complexity. The Supplementary Fig. 22 where the RA pre-weighting is used shows that LE is faster than the other methods and suggests that the computational complexity might be in the same range of RA-ncMCE or even lower, certainly not higher. Therefore, we conclude that the complexity would be $O(EN)$ if the EBC pre-weighting is used; and it could be even lower, and approximatively $O(N^2)$ for LE, MCE, and ncMCE, if the RA pre-weighting is used.

**Hardware and software details**. Unless stated otherwise MATLAB code was used for all the methods and simulations, which were carried out on a workstation under Windows 8.1 Pro with 512 GB of RAM and 2 Intel(R) Xenon(R) CPU E5-2687W v3 processors with 3.10 GHz.

**Code availability**. The MATLAB code for performing the coalescent embedding in 2D and 3D, together with functions for the evaluation (HD-correlation, C-score, and GR-score) and visualization of the embedding are publicly available at the GitHub repository: https://github.com/biomedical-cybernetics/coalescent_embedding.

**Data availability**. The data that support the findings of this study are available from the corresponding author on reasonable request. For real networks data that have been obtained from publicly available sources, the corresponding URLs or references are provided in the "Real networks data set" section.

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

## Acknowledgements

We thank the BIOTEC System Administrators for their IT support, Claudia Matthes for the administrative assistance, and the Centre for Information Services and High Performance Computing (ZIH) of the TUD. We are particularly grateful to Fragkiskos Papadopoulos for the valuable suggestions and for sharing the AS Internet data; to Dmitri Krioukov for the enlightening theoretical discussion on the relation between scale freeness and hyperbolicity; as well to M. Ángeles Serrano, Marian Boguña, and Maksim Kitsak for the useful discussions on this topic. We thank the MathWorks Support and Josef Schicker for the detailed information about the computational complexity of the MATLAB functions for matrix factorization. C.V.C. thanks Trey Ideker for introducing him (during a 1-year visiting scholar period at UCSD in 2009) to the problem of protein–protein interactions network embedding. C.V.C. thanks Roded Sharan, Natasa Przulj, and Timothy Ravasi for supporting the idea of using minimum curvilinearity for prediction of protein–protein interactions. C.V.C. thanks Franco Maria Montevecchi and Massimo Alessio for their unique mentoring during the period of PhD in which he developed minimum curvilinearity machine learning strategy. C.V.C. thanks Carlo Rovelli and Enzo Di Fabrizio for their kind feedback on abstract. The work in C.V.C. research group was mainly supported by: the independent group leader starting grant of

the Technische Universität Dresden (TUD); by the Klaus Tschira Stiftung (KTS) gGmbH, Germany (Grant number: 00.285.2016); by the research application pool (Forschungspoolantrags) of TUD. S.C. PhD fellowship is supported by Lipotype GmbH. A.M. was partially supported by the funding provided by the Free State of Saxony in accordance with the Saxon Scholarship Program Regulation, awarded by the Studentenwerk Dresden based on the recommendation of the board of the Graduate Academy of TU Dresden. We also acknowledge support by the Centre for Information Services and High Performance Computing (ZIH) of the TUD.

## Author contributions

C.V.C. envisaged the study, invented and designed the coalescent embedding, the pre-weighting and equidistant adjustment strategies, the algorithms, and the experiments. G.B. suggested to adopt the EBC measure as a pre-weighting option. A.M., J.M.T. and S.C. implemented and ran the codes and performed the computational analysis with C.V.C. help. All the authors analyzed the results. A.M., J.M.T. and S.C. realized the figures under the C.V.C. guidance. C.V.C. and A.M. wrote the article with input and corrections from all the other authors. C.V.C. led, directed, and supervised the study.

## Additional information

**Competing interests:** The authors declare no competing financial interests.

