## [Peer Review File · Nature Communications]

Reviewers' comments:

Reviewer #1 (Remarks to the Author):

Report on the manuscript titled "Machine learning meets complex networks: coalescent embedding in the hyperbolic space boosts community detection in real networks" by J. M. Thomas et al.

Network geometry is one of the promising approaches towards understanding structure and function of networked systems. Within the approach network nodes correspond to points in the underlying space such that connections between the nodes are determined by distances between corresponding points. Present manuscript is addressing a very challenging problem — mapping of real networks to underlying geometric spaces. It is hard to overestimate the importance of mapping accuracy since incorrectly or not accurately inferred node coordinates in real network inevitably affect all subsequent steps in the analysis of network geometry.

Currently available network mapping techniques (Refs. 2 and 16 in present manuscript) are developed and optimized for non-weighted networks without pronounced community structure and are quite slow ($O(N^3)$ - $O(N^4)$), where N is the network size. This severely limits their applicability to real systems.

In present manuscript authors optimize machine-learning algorithms for the purpose of mapping real networks to 2D hyperbolic spaces and demonstrate that their methods significantly outperform existing state of the art methods. I find the manuscript well written and adequately explained.

I believe that the results presented in this manuscript are potentially of great importance for many disciplines including systems biology, computer science, statistical physics and cosmology.

In summary, I believe the manuscript with proper revisions should be publishable in Nature Communications.

I have a number of conceptual and technical questions that I list below.

Conceptual Questions:

1. It seems that the idea of coalescent embedding is specific to the 2-dimensional space (1-dimensional angular space). At least it is not immediately clear how to generalize it to spaces of higher dimensionality. I would like to invite authors to discuss this aspect in the manuscript.

2. Related to my first question, I am curious if developed techniques are in principle applicable to latent spaces, other than hyperbolic disk. In particular, in doi:10.1038/srep00793 Krioukov et al argue that dual to Riemannian manifolds, is network representation in Lorentzian manifolds (e.g. deSitter spacetime).

3) In all considered algorithms, authors impose an "equidistant adjustment" of angular coordinates. This procedure is supposed to work well on synthetic networks since original angular distributions are uniform. In real networks, however, angular distances may be non-uniform. Thus, I am concerned that the "equidistant adjustment" ultimately leads to the loss of information. For instance, regions in the map of a real network with high node densities could signal the presence of communities. Also, "equidistant adjustment" may affect other important network characteristics like navigability. As a referee, I invite authors to discuss possible implications of the "equidistant adjustment" in the main text.

Technical Comments:

4) I note that all experiments showing alignment/agreement between inferred and real angles are for temperature $T=0$ (e.g., Fig. 4 in main text, Figs. 5, 6 in SI).

Since many real networks are characterized by $T>0$, I believe that authors need to include $T>0$ results for fair comparison, e.g. $T=0.5$.

5) Many machine learning methods in order to boost-up the running time use only the information about the connected nodes and do not take into account information about pairs of nodes that are disconnected. As a referee, I am wondering if this is the case here.

6) Related to my previous question, success ratio of greedy geometric routing is another important metric to assess the quality of the mapping. I invite authors to compare mappings using this metric as well (for $T>0$), as it is very sensitive to whether or not the absence of links is taken into account by the mapping method.

7) While authors do compare performance of their methods with existing ones, I would like to encourage them to do it in more systematic fashion. One way could be calculating (or estimating numerically) how the execution time scales with network size.

Other comments

8) It is not readily clear what meaning authors put into "coalescent" throughout the text. I would like to invite authors to consider a synonym or add a brief clarification for their choice of this term.

9) In the legend of Fig.4 authors say "For each subplot the value of HyperMap-CN for $T = 0$ is missing due to a performance decrease out of the plotting range, these values can be found in Suppl. Fig. 4. The methods without EA are plotted in Suppl. Fig. 3."

I am curious if the authors are using the original HyperMap-CN code (https://bitbucket.org/dk-lab/2015_code_hypermap) or they developed their own version of it?

In the case authors used their version of the HyperMap-CN code (or are using a modified version of it), they need to acknowledge this in the manuscript.

The original HyperMap-CN code assumes $T > 0$ (i.e., the code does not consider the limiting case $T=0$). So, what authors are observing for $T=0$ is probably the result of numerical overflows.

Authors need to adjust their interpretation accordingly.

3) Finally, I note that authors promised to release their code for public use. Yet, the code is currently not available.

I would therefore like to encourage author to release their code as soon as possible. Without the available mapping code, practical significance of the manuscript would be quite low.

Reviewer #2 (Remarks to the Author):

Authors describe a novel and exciting class of algorithms that offers fast and accurate embedding of even large graphs in the hyperbolic space, and can be used in network community detection.

Major comments:

1. Authors should reconcile their model of repulsion attraction rule with the existence of rich clubs, where hubs do not repulse each other at all, but just conversely form a cluster. The rich club

phenomenon is pretty usual in certain classes of networks while it is not that much present in other classes (see <http://www.nature.com/nphys/journal/v2/n2/full/nphys209.html>). Authors should try their method by comparing key examples of rich-club versus non-rich-club networks and should show whether they differ or not.

2. Authors should cite and discuss the following former papers on network hyperbolicity:

<https://arxiv.org/abs/1307.0031>

<https://arxiv.org/abs/1503.03061>

<https://arxiv.org/abs/1605.03059>

<http://arxiv.org/abs/1606.09481>

<http://arxiv.org/abs/1309.0459>

<http://arxiv.org/abs/1310.0761>

<http://arxiv.org/abs/cond-mat/0408443>

<http://arxiv.org/abs/0911.2538>

Minor comments:

1. please rephrase "seem specularly connected" in the abstract to "seem connected"
2. please rephrase "to boost the state of art community detection" to "to help community detection"
3. please write out PSO model in full when first mentioning in the Intro
4. please rewrite in the intro: "The node similarities are related" to "On one hand, node similarities are related"
5. Please refer to the original papers in the Results when speaking about networks like "Karate club", "Opsahl" etc.
6. Please rewrite these sections of Methods: "were theoretically designed by Cannistraci" to "were theoretically designed by Cannistraci et al" and cite papers 9 and 12, as well as "Hence, according to Cannistraci" to "Hence, according to Cannistraci et al" and cite here also papers 9 and 12.
7. References 9,10,11,12,20,25,27 and 31 are incomplete. Please complete them.

Reviewer #3 (Remarks to the Author):

This manuscript showed that the topological-based machine learning algorithms, which are often used for nonlinear dimensionality reduction, can be used to directly the node's angular coordinates of the hyperbolic model. As the main application, this has been used to weight the network such that community detection algorithms work better than the original unweighted networks. Although the work is interesting, it does not warrant publication in Nature Communications (see comments below). I recommend the authors revised the paper and resubmit it to a more technical machine learning venue (e.g. Machine Learning, Machine Learning Research or top conference such as NIPS, ICML and KDD).

Major comments:

- They should better discuss why the outcomes of the algorithms in Fig. 1 are unexpected.
- Other than the community identification, there is no major application for the proposed (computationally expensive) algorithms. Real-world applications should be included.
- The term "structural network imaging" should be used carefully, as it is, in its current format, can be misleading.
- The logic behind the choice of formulation for RA (Fig. 2) is not discussed. For example, when high degree nodes are considered (e.g., hubs in scale-free networks), what's the reason to include $d_i + d_j + d_i d_j$, as $d_i d_j$ will be far higher than d_i and d_j . This is also the case for those with small degrees; there is no that much difference between d_i , d_j and $d_i d_j$. It seems that this is just a random choice without any rigorous analysis. Without such a rigorous mathematical analysis, it is hard to accept that this is an optimal choice.
- The colour of lines in Figs 3 and 4 is hardly distinguishable; this should be corrected.
- No sophisticated analysis on the computational complexity of the algorithm has been given. They should give the mathematical analysis (e.g., $O(\dots)$) on the complexity of the algorithms, rather

than simply providing the running time, which can be influenced by many factors in the time of numerical simulations.

- At the end of the day, the main product of this work is a new algorithm for community identification. However, the authors have just showed they have improved the performance of the baseline algorithms. There is a rich literature on the community identification algorithms (in both computer science and physics communities), and many state-of-the-art algorithms work better than these baseline algorithms (some of existing algorithms also work by first weighting the network, and then applying the standard algorithm). The proposed algorithm must be compared with the state-of-the-art ones.

- In the revision, please do not consider small-scale networks such as Karate Club, please !!!

Point-by-point reply to reviewers' comments

Dear reviewers, to simplify your review, all the major modifications generated to address your comments are reported in red characters in the new text of the draft.

The novel draft changed the title to address a comment of the third reviewer, and was enriched with new important sections and several comments along the text aimed to address your remarks by including also new results and the related methods. However, since it was difficult to integrate in the present draft new proofs of the immediate impact of our methodology on other disciplines, we also produced a short and focused back-to-back draft that deals with the latent geometry of brain connectomes. The title of the draft is *Coalescent embedding in the hyperbolic space unsupervisedly discloses the hidden geometry of the brain* and it is submitted to Nature Communication. We hope that the journal will consider to send it for review to you, together with the present revised manuscript.

Reviewer #1

Report on the manuscript titled “Machine learning meets complex networks: coalescent embedding in the hyperbolic space boosts community detection in real networks” by J. M. Thomas et al.

Network geometry is one of the promising approaches towards understanding structure and function of networked systems. Within the approach network nodes correspond to points in the underlying space such that connections between the nodes are determined by distances between corresponding points. Present manuscript is addressing a very challenging problem — mapping of real networks to underlying geometric spaces. It is hard to overestimate the importance of mapping accuracy since incorrectly or not accurately inferred node coordinates in real network inevitably affect all subsequent steps in the analysis of network geometry.

Currently available network mapping techniques (Refs. 2 and 16 in present manuscript) are developed and optimized for non-weighted networks without pronounced community structure and are quite slow ($O(N^3)$ - $O(N^4)$), where N is the network size. This severely limits their applicability to real systems.

In present manuscript authors optimize machine-learning algorithms for the purpose of mapping real networks to 2D hyperbolic spaces and demonstrate that their methods significantly outperform existing state of the art methods. I find the manuscript well written and adequately explained.

I believe that the results presented in this manuscript are potentially of great importance for many disciplines including systems biology, computer science, statistical physics and cosmology.

In summary, I believe the manuscript with proper revisions should be publishable in Nature Communications.

Reply: we thank the reviewer for the interest and consideration showed for the work presented in our draft. As the reviewer emphasized above, our method can be of importance for many disciplines. Hence, in order to give proof of the immediate impact of our methodology, we also produced a short and focused back-to-back draft that deals with the latent geometry of brain connectomes. The title of the article is *Coalescent embedding in the hyperbolic space unsupervisedly discloses the hidden geometry of the brain* and we hope it will be provided for review together with the present revised manuscript.

I have a number of conceptual and technical questions that I list below.

Conceptual Questions:

1.1. It seems that the idea of coalescent embedding is specific to the 2-dimensional space (1-dimensional angular space). At least it is not immediately clear how to generalize it to spaces of higher dimensionality. I would like to invite authors to discuss this aspect in the manuscript.

Reply: we are delighted to receive this interesting hint. We investigated how some of the coalescent embedding techniques can be extended to a space of higher dimensionality, in particular the 3D hyperbolic sphere. Our quantitative evaluations on greedy routing and community detection, both in real and artificial networks, show that overall there is not a significant improvement of performance with the introduction of an additional dimension. We introduced a dedicated section in the Results named "Beyond the two-dimensional space" in the article, in which we extensively discuss this point.

1.2. Related to my first question, I am curious if developed techniques are in principle applicable to latent spaces, other than hyperbolic disk. In particular, in doi:10.1038/srep00793 Krioukov et al argue that dual to Riemannian manifolds, is network representation in Lorentzian manifolds (e.g. deSitter spacetime).

Reply: since we use techniques of topological-based nonlinear dimension reduction, in principle they should be applicable to any latent space. The only limitation consists in the ability to learn a kernel that approximates the geometry of the hidden manifold behind the network topology. At the moment we admit that we are not able to provide a solution to the question of the reviewer, and that only further scientific efforts will help us in the years to come to take advantage of the technique proposed in this article to address many other questions in network geometry. For instance, we could obtain a network representation in Lorentzian manifolds (deSitter space) just by applying the mapping between the Poincare' disk and the de Sitter space proposed in doi:10.1038/srep00793 Krioukov et al. However, we believe that this is only an initial consideration towards the direction pointed by the reviewer.

1.3. In all considered algorithms, authors impose an "equidistant adjustment" of angular coordinates. This procedure is supposed to work well on synthetic networks since original angular distributions are uniform. In real networks, however, angular distances may be non-uniform. Thus, I am concerned that the "equidistant adjustment" ultimately leads to the loss of information. For instance, regions in the map of a real network with high node densities could signal the presence of communities. Also, "equidistant adjustment" may affect other important network characteristics like navigability. As a referee, I invite authors to discuss possible implications of the "equidistant adjustment" in the main text.

Reply: we thank again the reviewer for the valuable suggestion. We performed the greedy routing both on real and synthetic networks, highlighting the comparison between methods with and without equidistant adjustment (Fig. 8, Suppl. Fig. 20-21). We introduced a dedicated paragraph for the possible implications of the equidistant adjustment, considering both evaluations on synthetic networks and applications on real networks in the section "*Results - Some notes on pre-weighting, rich-clubness and angular adjustment*".

Technical Comments:

1.4 I note that all experiments showing alignment/agreement between inferred and real angles are for temperature $T=0$ (e.g., Fig. 4 in main text, Figs. 5, 6 in SI).

Since many real networks are characterized by $T>0$, I believe that authors need to include $T>0$ results for fair comparison, e.g. $T=0.5$.

Reply: the scatterplots (Fig. 4A) have mainly been reported for a qualitative visualisation of the agreement between the original and the inferred-aligned angular coordinates. Instead, the quantitative evaluation has been performed using the C-score, for which all the parameter combinations are shown (Fig. 4B). However, for sake of completeness, we have now introduced the scatterplots for all the methods and all the values of temperature considered, $T = [0, 0.3, 0.6, 0.9]$, as supplementary information (Suppl. Fig. 10-19).

1.5. Many machine learning methods in order to boost-up the running time use only the information about the connected nodes and do not take into account information about pairs of nodes that are disconnected. As a referee, I am wondering if this is the case here.

Reply: we will begin the answer to this elegant comment of the reviewer citing an entire paragraph of a landmark article on nonlinear dimension reduction by Saul and Roweis [1].

<< Note that while in Local Linear Embedding (LLE) the reconstruction weights for each data point are computed from its local neighbourhood - independent of the weights for other data points - the embedding coordinates are computed by an $N \times N$ eigensolver, a global operation that couples all data points (or more precisely, all data points that lie in the same connected component of the graph defined by the neighbors). This is how the algorithm discovers global structure - by integrating information from overlapping local neighborhoods. >>

In practice, all the algorithms that we use are global methods because they exploit an $N \times N$ matrix decomposition. However, Laplacian Eigenmaps (LE) is a neighbourhood-preserving global methods like LLE, in fact, the Laplacian matrix to which eigen-decomposition is applied only contains information about connected nodes. Isomap (ISO) and its non-centered variants (ncISO), on the contrary, belong to the class of global methods that aim to preserve the global-topology of the neighbourhood graph that approximates the hidden manifold geometry. Meaning that they attempt to preserve geometry at all scales, mapping nearby points on the manifold to nearby points in low-dimensional space, and faraway points to faraway points. Indeed, they apply singular value decomposition to a distance kernel, which contains information about both connected and disconnected nodes.

Minimum Curvilinear Embedding (MCE) and its non-centered variants (ncMCE), are global methods that preserve a locally-reconstructed (by means of MST) global-geometry. Since the MC kernel is obtained by computing all pairwise transversal distances over the MST, in practice distances for both connected and disconnected nodes are preserved.

However, the general computational complexity of all these techniques is $O(N^2)$ for LE, ncMCE and MCE, and $O(EN)$ for ISO and ncISO. LE uses only the connected nodes information, hence it is a bit faster because solves a sparse eigenvalue problem, but it still provides in general at least the same computational complexity of ncMCE and MCE. See Suppl. Fig. 25 for numerical time assessment and the dedicated section in the Methods for computational complexity.

[1] Saul, L. K. L. & Roweis, S. S. T. *Think globally, fit locally: unsupervised learning of low dimensional manifolds. J. Mach. Learn. Res. 4, 119–155 (2003).*

1.6. Related to my previous question, success ratio of greedy geometric routing is another important metric to assess the quality of the mapping. I invite authors to compare mappings using this metric as well (for $T > 0$), as it is very sensitive to whether or not the absence of links is taken into account by the mapping method.

Reply: as suggested by the reviewer, we have added the evaluation of the greedy routing both on synthetic networks (Fig. 8 and Suppl. Fig. 20, discussed in “*Results - Greedy routing performance in synthetic and real networks*”). In general, we found that MCE-based techniques are the coalescent embedding techniques that performs better on both synthetic and real networks

1.7. While authors do compare performance of their methods with existing ones, I would like to encourage them to do it in more systematic fashion. One way could be calculating (or estimating numerically) how the execution time scales with network size.

Reply: we thank the reviewer for the suggestion. We introduced a new section, “*Methods - Complexity of coalescent embedding algorithms*” in the article, which is entirely dedicated to the detailed analysis of the computational complexity. Furthermore, we have included an additional plot (Suppl. Fig. 25) that highlights how the computational time of the different methods scales for networks of increasing size, going from 100 up to 30000 nodes. In general, all coalescent embedding methods display a computational time that is in the same temporal range, with methods like LE that are based on sparse decomposition that performs slightly better.

Other comments

1.8. It is not readily clear what meaning authors put into “coalescent” throughout the text. I would like to invite authors to consider a synonym or add a brief clarification for their choice of this term.

Reply: we have now added in the text a proper explanation for the choice of the term ‘coalescent’.

1.9. In the legend of Fig.4 authors say “For each subplot the value of HyperMap-CN for $T = 0$ is missing due to a performance decrease out of the plotting range, these values can be found in Suppl. Fig. 4. The methods without EA are plotted in Suppl. Fig. 3.”

I am curious if the authors are using the original HyperMap-CN code (https://bitbucket.org/dk-lab/2015_code_hypermap) or they developed their own version of it?

In the case authors used their version of the HyperMap-CN code (or are using a modified version of it), they need to acknowledge this in the manuscript.

The original HyperMap-CN code assumes $T > 0$ (i.e., the code does not consider the limiting case $T=0$). So, what authors are observing for $T=0$ is probably the result of numerical overflows. Authors need to adjust their interpretation accordingly.

Reply: thanks for pointing this out. We are using the code taken from the website (https://bitbucket.org/dk-lab/2015_code_hypermap). Since, as you are suggesting, there can be a numerical overflow in the limiting case $T = 0$, this is probably the reason of the performance decrease. Therefore, we have removed the HyperMap-CN performance for $T = 0$ in the plots and we have reported the explanation in the figure legend.

1.10. Finally, I note that authors promised to release their code for public use. Yet, the code is currently not available.

I would therefore like to encourage author to release their code as soon as possible. Without the available mapping code, practical significance of the manuscript would be quite low.

Reply: as soon as the article will be accepted the code will be publicly released on: <https://github.com/biomedical-cybernetics>

Reviewer #2 (Remarks to the Author):

Authors describe a novel and exciting class of algorithms that offers fast and accurate embedding of even large graphs in the hyperbolic space, and can be used in network community detection.

Major comments:

2.1. Authors should reconcile their model of repulsion attraction rule with the existence of rich clubs, where hubs do not repulse each other at all, but just conversely form a cluster. The rich club phenomenon is pretty usual in certain classes of networks while it is not that much present in other classes (see <http://www.nature.com/nphys/journal/v2/n2/full/nphys209.html>). Authors should try their method by comparing key examples of rich-club versus non-rich-club networks and should show whether they differ or not.

Reply: we thank the reviewer for this very interesting hint. In rich-club networks, high degree nodes (hubs) tend to connect each other. We would like to clarify that the repulsive part of the rule is not suggesting that nodes with high degree tend to be disconnected. It suggests that they tend to dominate geometrically distant regions, which does not exclude their connectivity and therefore it should not be theoretically in contrast to the existence of rich-clubs. However, we agreed to prove this point by experiments and we found the comparison between rich-club and non-rich-club networks interesting. Since a procedure that univocally assigns a significance value for rich-clubness to a given network was still missing, we conducted an entire study proposing a new statistical test for rich-clubness together with a new congruous null model. The related article is already available on arXiv at [1]. We performed the statistical test on the PSO networks, the p-values are reported in Suppl. Fig. 24. The statistical test highlights that for most of the parameter combinations, in particular for $m = [4, 6]$ and $N = [500, 1000]$ the networks present a significant rich-club, whereas for more sparse ($m = 2$) and small networks ($N = 100$) in general there is not a significant rich-club. This is in agreement with the network growing procedure explained by the PSO model. In fact, the high degree nodes are the first ones to be born in the network and they are expected to connect to around m of the older nodes. Therefore for higher m the

rich nodes have higher probability to create a club. Looking at Suppl. Fig. 4-8, it is evident that for networks with $m = [4, 6]$, containing a significant rich-club, the methods using the RA pre-weighting rule do not have any particular decrease in performance, they still provide a very high improvement with respect to the unweighted variant, as for networks with $m = 2$ that do not contain a significant rich-club. We therefore conclude that the RA pre-weighting rule can be adopted regardless of the rich-clubness of a network. We have introduced this discussion in the “Results - Some notes on pre-weighting, rich-clubness and angular adjustment” section in the text.

[1] Muscoloni, A. & Cannistraci, C. V. Rich-clubness test: how to determine whether a complex network has or doesn't have a rich-club? *arXiv:1704.03526v1 [physics.soc-ph]* (2017).

2.2. Authors should cite and discuss the following former papers on network hyperbolicity:

<https://arxiv.org/abs/1307.0031>

<https://arxiv.org/abs/1503.03061>

<https://arxiv.org/abs/1605.03059>

<http://arxiv.org/abs/1606.09481>

<http://arxiv.org/abs/1309.0459>

<http://arxiv.org/abs/1310.0761>

<http://arxiv.org/abs/cond-mat/0408443>

<http://arxiv.org/abs/0911.2538>

Reply: we thank the reviewer for pointing out this relevant literature. We have now added a discussion of these papers in the resubmitted version of our manuscript.

Minor comments:

2.3. please rephrase "seem specularly connected" in the abstract to "seem connected"

2.4. please rephrase "to boost the state of art community detection" to "to help community detection"

2.5. please write out PSO model in full when first mentioning in the Intro

2.6. please rewrite in the intro: "The node similarities are related" to "On one hand, node similarities are related"

2.7. Please refer to the original papers in the Results when speaking about networks like "Karate club", "Opsahl" etc.

2.8. Please rewrite these sections of Methods: "were theoretically designed by Cannistraci" to "were theoretically designed by Cannistraci et al" and cite papers 9 and 12, as well as "Hence, according to Cannistraci" to "Hence, according to Cannistraci et al" and cite here also papers 9 and 12.

2.9. References 9,10,11,12,20,25,27 and 31 are incomplete. Please complete them.

Reply: thanks, we have modified the manuscript's title and text accordingly.

Reviewer #3 (Remarks to the Author):

This manuscript showed that the topological-based machine learning algorithms, which are often used for nonlinear dimensionality reduction, can be used to directly the node's angular coordinates of the hyperbolic model. As the main application, this has been used to weight the network such that community detection algorithms work better than the original unweighted networks.

Although the work is interesting, it does not warrant publication in Nature Communications (see comments below). I recommend the authors revised the paper and resubmit it to a more technical machine learning venue (e.g. Machine Learning, Machine Learning Research or top conference such as NIPS, ICML and KDD).

Reply: dear reviewer, please notice that even the first reviewer stated: "I believe that the results presented in this manuscript are potentially of great importance for many disciplines including systems biology, computer science, statistical physics and cosmology."

We agree with you that the first version of the draft did not sufficiently discuss the impact on other disciplines. However, as the first reviewer emphasized above, our method can be of importance for many disciplines and it was difficult to add further examples and applications in the present draft. To convince you that our draft deserves more attention than a more technical machine learning venue, and since the new mole of information was too big to be included in the present manuscript, we were obliged to design a back-to-back draft.

This short and focused back-to-back draft is proposed in order to give proof of the immediate impact of our methodology, and deals with the latent geometry of brain connectomes. The title of the article is *Coalescent embedding in the hyperbolic space unsupervisedly discloses the hidden geometry of the brain* and we hope that the journal will agree to send you for review together with the present revised manuscript.

Major comments:

3.1. They should better discuss why the outcomes of the algorithms in Fig. 1 are unexpected.

Reply: we thank the reviewer for spotting this point. We actually meant something different from what the reviewer commented. In order to avoid confusion, we removed the word unexpected from the text.

3.2. Other than the community identification, there is no major application for the proposed (computationally expensive) algorithms. Real-world applications should be included.

Reply: we thank the reviewer for the suggestion. Actually, we have other interesting real-world applications both on brain networks and molecular networks, but due to the present revision the article has become already quite extensive and we would not like to overload it with many applications, but rather focus on the introduction of the new methods and their test. We prefer to present and discuss the results of further applications in separate articles, one of which (on brain connectomes) is now provided to the reviewer as back-to-back manuscript for his review, and in order to convince him of the value of the proposed methodology.

Finally, the reviewer defined our class of proposed algorithms computational expensive. We kindly disagree because the complexity of our algorithm is $O(N^2)$ for LE, ncMCE and MCE and $O(EN)$ for ISO and

ncISO; and as showed in the new additional plot (Suppl. Fig. 25, that highlights how the computational time of the different methods scales for networks of increasing size, going from 100 up to 30.000 nodes) embedding of networks of 30.000 nodes can be efficiently performed in less than 10 minutes as worst performance and 1 minutes as best performance.

3.3. The term “structural network imaging” should be used carefully, as it is, in its current format, can be misleading.

Reply: Thanks for the suggestion. Actually, we forgot to use the quotation marks. We now corrected the expression using the quotation marks. In addition, since we knew that the term could have been misleading, we described our interpretation right after the introduction of the term.

3.4. The logic behind the choice of formulation for RA (Fig. 2) is not discussed. For example, when high degree nodes are considered (e.g., hubs in scale-free networks), what’s the reason to include $d_i + d_j + d_i d_j$, as $d_i d_j$ will be far higher than d_i and d_j . This is also the case for those with small degrees; there is no that much difference between d_i , d_j and $d_i d_j$. It seems that this is just a random choice without any rigorous analysis. Without such a rigorous mathematical analysis, it is hard to accept that this is an optimal choice.

Reply: we apologize if we were too concise with the introduction of the RA formula. We thank the reviewer for the comment that gave us the opportunity to improve this aspect of the draft. We designed four different variants of the RA formula, differing in the way in which the degrees of the connected nodes are combined at the numerator (the repulsive part of the rule). The performances obtained with the four different pre-weighting mathematical expressions of the same rule are summarized in Suppl. Fig. 23, both for small-size and large-size synthetic networks. The results are discussed in the “*Results - Some notes on pre-weighting, rich-clubness and angular adjustment*” section and they highlight that the difference in performance is negligible. We therefore propose to adopt the two variants that overall offered the best scores respectively for small-size (RA_1) and large-size (RA_2) networks, whose mathematical expressions are also reported in Fig. 2. However, for sake of brevity, since for most of the simulations the networks are small and the results are very close, we show all the other results only for RA_1 .

3.5. The colour of lines in Figs 3 and 4 is hardly distinguishable; this should be corrected.

Reply: we agree with the reviewer that the performance of the different versions of each algorithm is hardly distinguishable, but this would not change also in the case that we use different colours. The figure is already full of multiple lines with different colours and we prefer to keep a similar colour with a different tone for the same version of each algorithm in order to emphasize the diversity between different algorithms and the similarity of performance for the same algorithm.

3.6. No sophisticated analysis on the computational complexity of the algorithm has been given. They should give the mathematical analysis (e.g., $O(\dots)$) on the complexity of the algorithms, rather than

simply providing the running time, which can be influenced by many factors in the time of numerical simulations.

Reply: we apologize if we did not make this important analysis in the first version of the draft. We agree with the reviewer that the analysis of computational complexity is incumbent. We have introduced in the new draft a new section, “Methods - Complexity of coalescent embedding algorithms”, entirely dedicated to the detailed analysis of the computational complexity. Furthermore, we have included an additional plot (Suppl. Fig. 25) that highlights how the computational time of the different methods scales for networks of increasing size, going from 100 up to 30000 nodes.

3.7. At the end of the day, the main product of this work is a new algorithm for community identification. However, the authors have just showed they have improved the performance of the baseline algorithms. There is a rich literature on the community identification algorithms (in both computer science and physics communities), and many state-of-the-art algorithms work better than these baseline algorithms (some of existing algorithms also work by first weighting the network, and then applying the standard algorithm). The proposed algorithm must be compared with the state-of-the-art ones.

Reply: At the end the day, the reviewer should have now (if the journal accepted our back-to-back draft submission) also for review a short and focused back-to-back draft on brain networks (in healthy and pathological conditions) that clearly demonstrates the impact of the new class of efficient algorithms proposed in the present draft. This is a proof that our new algorithms do not exclusively impact community detection, but can impact any discipline that is interested to mine the latent geometry of complex networks.

Furthermore, we thank the reviewer for the useful comment, since it revealed the need to stress in a stronger way what is intended to be the main product of the work and the real message that we want to deliver with this article. In order to avoid the possibility of misunderstandings we removed from the title of the article the term community detection and any direct reference to this task.

The investigation of the hidden geometry behind complex network topologies is an active research topic in recent years, and the PSO model highlighted that the hyperbolic space can offer an adequate representation of the network geometry in a low dimensional space. However, in absence of a method able to accurately map the network in a reasonable computational time, any further analysis in the geometrical space would be compromised. Here we propose a new class of machine learning algorithms that offer a fast and accurate embedding in the hyperbolic space even for large graphs. This is intended to be the main product of the work. Afterward, several studies can be lead exploiting the geometrical information, covering disparate application fields like social, communication, transportation, biological and brain networks. Nevertheless, in order to not overload the article, we presented only one main application showing how the hyperbolic distances can be used to feed community detection algorithms. We changed the text in order to make more clear what is the main innovation proposed and its wide range of applicability, not certainly limited to community detection.

Anyway, for making the community detection study more complete, we have introduced two additional state-of-the-art algorithms. Yang et al. [1] recently performed a comparative study of 8 community detection methods. The comparison highlights that overall, taking into consideration both the

performance and the computational time, the four best methods are Louvain, Infomap, Label propagation and Walktrap. Since two of them (Louvain and Infomap) were already present in our analysis, we decided to add Label propagation and Walktrap, which can also take in input weighted adjacency matrices.

[1] Yang, Z., Algesheimer, R., & Tessone, C. J. (2016). A Comparative Analysis of Community Detection Algorithms on Artificial Networks. *Scientific Reports*, 6, 30750.

3.8. In the revision, please do not consider small-scale networks such as Karate Club, please!!!

Reply: we agree that also large-size networks should be included, we have therefore introduced Internet networks up to 37000 nodes both for community detection and greedy routing.

REVIEWERS' COMMENTS:

Reviewer #1 (Remarks to the Author):

Report on the manuscript titled "Machine learning meets complex networks: coalescent embedding in the hyperbolic space boosts community detection in real networks" by J. M. Thomas et al.

The revised manuscript addressed all my original questions and now is in principle publishable in Nature Communications.

I have two follow-up comments for the authors.

1. In note that authors explored the possibility of a 3D mapping of synthetic networks generated in 2D hyperbolic spaces, reporting little to no benefit of using the extra dimension. While this is indeed, expected, true benefits may arise when the true latent space is also 3-dimensional. Therefore, an interesting test could be to generate synthetic using PSO in 3D and then to compare the accuracy of embedding using 3D and 2D mappings. This is suggestion is fully optional and could be explored in a separate work.

2. I am curious what techniques authors used to perform equidistant adjustment on the sphere. Somehow, I could not find the technical details in the manuscript. In the case they are indeed included, I would invite authors to provide a reference to them from the legend of Fig. 8.

Reviewer #2 (Remarks to the Author):

Authors have addressed all of my comments adequately.

Reviewer #3 (Remarks to the Author):

The authors have significantly improved the manuscript although partly (I should say cleverly!) ignored my original comments. Although I am still against supporting publication of the current version, I enjoyed reading their their companion paper on brain networks. I suggest to combine the two papers and have the other paper as only a (significant) application of the proposed methodology. This makes the work much stronger.

Reviewer #1

The revised manuscript addressed all my original questions and now is in principle publishable in Nature Communications.

I have two follow-up comments for the authors.

1. In note that authors explored the possibility of a 3D mapping of synthetic networks generated in 2D hyperbolic spaces, reporting little to no benefit of using the extra dimension. While this is indeed, expected, true benefits may arise when the true latent space is also 3-dimensional. Therefore, an interesting test could be to generate synthetic using PSO in 3D and then to compare the accuracy of embedding using 3D and 2D mappings. This is suggestion is fully optional and could be explored in a separate work.

Reply: we thank the reviewer for the interesting suggestion. Actually, although it is expected that the addition of the third dimension does not lead to a significant benefit in the embedding of a 2D PSO model, we obtain the same result also in real networks, for which the hidden geometry is not necessarily 2D. However, the comparison of the accuracy in the embedding of a 2D versus 3D PSO model would be certainly an important point of investigation, but we prefer to leave it for a separate work. Nevertheless, we believe that the comment of the reviewer is important, and we have now included in the draft a part of the text that takes care to address the congruous observation raised by the reviewer. For simplicity, we report the text here below:

<< Overall, the tests both on real and artificial networks represent a quantitative evidence that the addition of the third dimension of embedding in the hyperbolic space does not lead to a clear and significant improvement in performance. Although for the PSO model this is indeed expected (because the synthetic networks are generated by a 2D geometrical model), we obtain the same result also on real networks, for which the hidden geometry is not necessarily 2D. Therefore, we might conclude that in practical applications, at least on the tested networks, the two-dimensional space appears to be enough for explaining the hidden geometry behind the complex network topologies. However, further investigations should be provided on networks of larger size and different types of origin, because the 3D space might conceptually offer an advantage with networks of large size. An additional interesting test can be to generate synthetic networks using a 3D PSO model, and then to compare the embedding accuracy using mapping techniques in 2D and 3D. >>

2. I am curious what techniques authors used to perform equidistant adjustment on the sphere. Somehow, I could not find the technical details in the manuscript. In the case they are indeed included, I would invite authors to provide a reference to them from the legend of Fig. 8.

Reply: we thank the reviewer for the comment. In the 3D mapping we do not perform the equidistant adjustment, since it is not trivial to find a meaningful way to equidistantly reorganize the pattern of similarities over a sphere. Furthermore, as suggested by the reviewer, we have now more explicitly indicated the techniques adopted for the 3D, both in the section ‘Beyond the two-dimensional space’ and in the Methods section ‘Coalescent embedding algorithm in 3D’.

Reviewer #3

The authors have significantly improved the manuscript although partly (I should say cleverly!) ignored my original comments. Although I am still against supporting publication of the current version, I enjoyed reading their companion paper on brain networks. I suggest to combine the two papers and have the other paper as only a (significant) application of the proposed methodology. This makes the work much stronger.

Reply: we thank the reviewer for having appreciated the improvements in the manuscript and for having enjoyed reading our new Cacciola et al. paper on brain networks, which certainly better shows a significant application of the proposed techniques. Although we understand that the application on brain networks is stronger than the one on community detection and greedy routing, we prefer to keep that application separated from this article, so that we can focus the first article on the introduction of the computational techniques for coalescent embedding. The second article requires a more detailed discussion from the network neuroscience point of view. We believe that merging the two manuscripts would overload too much the current draft, without any benefit for the reader, who can refer to the second article for further investigation on the impact of the algorithm in other fields such as computational neuroscience.